# HarmoniCa: Harmonizing Training and Inference for Better Feature Cache in Diffusion Transformer Acceleration

*"A tranquil forest clearing bathed in soft, magical light, filled with fairies dancing among the flowers. The pastel chalk drawing style gives the image a delicate, almost ethereal quality, with soft, smudged edges and gentle, powdery colors blending seamlessly."*

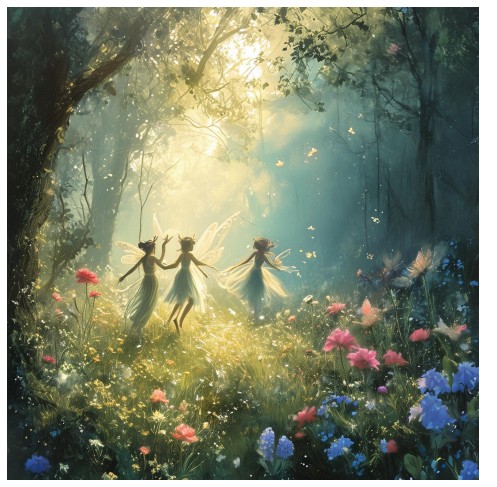 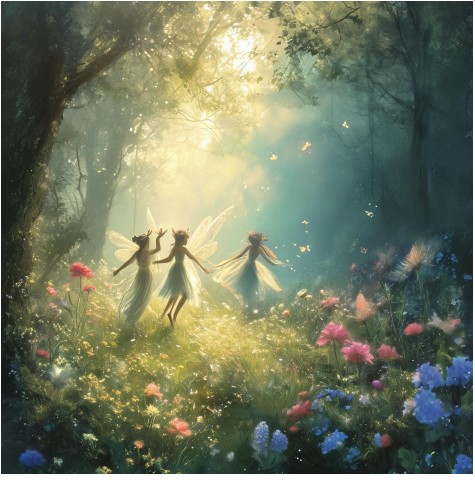

(a) PIXART-Σ *w/o* feature cache        (b) HarmoniCa (×1.68)

Figure 1: High-resolution $2048 \times 2048$ images generated using PIXART-Σ (Chen et al., 2024a) with a 20-step DPM-Solver++ sampler (Lu et al., 2022b). Our proposed feature cache framework achieves a substantial ×1.68 speedup. More visualization results can be found in Sec. T.

## ABSTRACT

Diffusion Transformers (DiTs) have gained prominence for outstanding scalability and extraordinary performance in generative tasks. However, their considerable inference costs impede practical deployment. The feature cache mechanism, which involves storing and retrieving redundant computations across timesteps, holds promise for reducing per-step inference time in diffusion models. Most existing caching methods for DiT are manually designed. Although the learning-based approach attempts to optimize strategies adaptively, it suffers from discrepancies [1] between training and inference, which hampers both the performance and acceleration ratio. Upon detailed analysis, we pinpoint that these discrepancies primarily stem from two aspects: (1) *Prior Timestep Disregard*, where training ignores the effect of cache usage at earlier timesteps, and (2) *Objective Mismatch*, where the training target (align predicted noise in each timestep) deviates from the goal of inference (generate the high-quality image). To alleviate these discrepancies, we propose **HarmoniCa**, a novel method that **harmoni**zes training and inference with a novel learning-based **ca**ching framework built upon *Step-Wise Denoising Training* (SDT) and *Image Error Proxy-Guided Objective* (IEPO). Compared

---

[1]In this paper, the discrepancy between training and inference denotes the mismatch or the inconsistency between these two processes.

to the traditional training paradigm, the newly proposed SDT maintains the continuity of the denoising process, enabling the model to leverage information from prior timesteps during training, similar to the way it operates during inference. Furthermore, we design IEPO, which integrates an efficient proxy mechanism to approximate the final image error caused by reusing the cached feature. Therefore, IEPO helps balance final image quality and cache utilization, resolving the issue of training that only considers the impact of cache usage on the predicted output at each timestep. Extensive experiments on class-conditional and text-to-image (T2I) tasks for 8 models and 4 samplers with resolutions ranging from $256 \times 256$ to $2048 \times 2048$ demonstrate the exceptional performance and speedup capabilities of our HarmoniCa. For example, HarmoniCa is the first feature cache method applied to the 20-step PIXART-$\alpha$ $1024 \times 1024$ that achieves over $1.5\times$ speedup in latency with an improved FID compared to the non-accelerated model. Remarkably, HarmoniCa requires no image data during training and reduces about 25% of training time compared to the existing learning-based approach.

# 1 INTRODUCTION

Diffusion models (Ho et al., 2020; Dhariwal & Nichol, 2021) have recently gained increasing popularity in a variety of generative tasks, such as image (Saharia et al., 2022; Esser et al., 2024) and video generation (Blattmann et al., 2023; Ma et al., 2024a), due to their ability to produce diverse and high-quality samples. Among different backbones, Diffusion Transformers (DiTs) (Peebles & Xie, 2023) stand out for offering exceptional scalability. However, the extensive parameter size and multi-round denoising nature of diffusion models bring tremendous computational overhead during inference, limiting their practical applications. For instance, generating one $2048 \times 2048$ resolution image using PixArt-$\Sigma$ (Chen et al., 2024a) with 0.6B parameters and 20 denoising rounds can take up to 14 seconds on a single NVIDIA H800 80GB GPU, which is unacceptable.

To accelerate the generation process of diffusion models, previous methods are developed from two perspectives: reducing the number of sampling steps (Liu et al., 2022; Song et al., 2020b) and decreasing the network complexity in noise prediction of each step (Fang et al., 2023; He et al., 2024). Recently, a new branch of research (Selvaraju et al., 2024; Yuan et al., 2024; Chen et al., 2024b) has started to focus on accelerating sampling time per step by the feature cache mechanism. This technique takes advantage of the repetitive computations across timesteps in diffusion models, allowing previously computed features to be cached and reused in later steps. Nevertheless, most existing methods are either tailored to the U-Net architecture (Ma et al., 2024c; Wimbauer et al., 2024) or develop their strategy based on empirical observations (Chen et al., 2024b; Selvaraju et al., 2024), and there is a lack of adaptive and systematic approaches for DiT models. Learning-to-Cache (Ma et al., 2024b) introduces a learnable router to guide the cache scheme for DiT models. However, this method induces discrepancies between training and inference, which always leads to distortion build-up (Ning et al., 2023; Li et al., 2024b; Ning et al., 2024). The discrepancies arise from two main factors: (1) *Prior Timestep Disregard*: During training, the model directly samples a timestep and employs the training images manually added noise akin to DDPM (Hu et al., 2021), ignoring the impact of the feature cache mechanism from earlier steps, which differs from the inference process. (2) *Objective Mismatch*: The training objective minimizes noise prediction error of each timestep, while the inference goal aims for high-quality final images, causing a misalignment in objectives. We believe these inconsistencies hinder effective and efficient router learning.

To alleviate the above discrepancies effectively, we present harmonizing training and inference with HarmoniCa, a novel cache learning framework featuring a unique training paradigm and a distinct learning objective. Specifically, to mitigate the first disparity, we design *Step-Wise Denoising Training* (SDT), which aligns the training process with the full denoising trajectory of inference using a student-teacher model setup. The student utilizes the cache while the teacher does not, effectively mimicking the teacher's outputs across all continuous timesteps. This approach maintains the reuse and update of the cache at earlier timesteps, similar to inference. Additionally, to address the misalignment in optimization goals, we introduce the *Image Error Proxy-Guided Objective* (IEPO), which leverages a proxy to approximate the final image error and reduces the significant costs of directly utilizing the error to supervise training. This objective helps SDT efficiently balance cache usage and image quality. By combining SDT and IEPO, extensive experiments for text-to-image

(T2I) and class-conditioned generation tasks show the promising performance and speedup ratio of HarmoniCa, *e.g.*, a $\times 1.51$ speedup and even a lower FID (Nash et al., 2021) for PIXART-$\alpha$ $1024 \times 1024$ (Chen et al., 2023). In addition, HarmoniCa eliminates the requirement of training with a large amount of image data and reduces about $25\%$ training time compared to the existing learning-based method (Ma et al., 2024b), further enhancing its applicability.

Our contributions are summarized as follows:

- We uncover two discrepancies between training and inference in the existing learning-based feature cache method: (1) *Prior Timestep Disregard*, indicating that the training process overlooks the influence of preceding timesteps, which is inconsistent with the inference process. (2) *Objective Mismatch*, minimizing intermediate outputs error, instead of the final image error. These discrepancies prevent the method from further performance and acceleration improvements.

- We propose a novel framework called HarmoniCa to alleviate the discovered discrepancies by: (1) *Step-Wise Denoising Training* (SDT), which addresses the first discrepancy by capturing the complete denoising trajectory, ensuring that the model learns to consider the impact of earlier timesteps. (2) *Image Error Proxy-Guided Optimization Objective* (IEPO), which mitigates the second discrepancy by using a proxy for the final image error, and thereby targets aligning the training objective with the inference.

- Extensive experiments on NVIDIA H800 80GB GPUs for DiT-XL/2, PIXART-$\alpha$, and PIXART-$\Sigma$ series–encompassing 8 models, 4 samplers, and 4 resolutions–proves the substantial efficacy and universality of HarmoniCa. For instance, it outperforms previous state-of-the-art (SOTA) by a 6.74 IS increase and 1.24 FID decrease with a higher speedup ratio on DiT-XL/2 $256 \times 256$. Notably, our image-free framework with much lower training cost exhibits superior efficiency and applicability than the current learning-based method.

## 2  RELATED WORK

**Diffusion models.** Diffusion models, initially conceptualized with the U-Net architecture (Ronneberger et al., 2015), have achieved satisfactory performance in image (Rombach et al., 2022; Podell et al., 2023) and video generation (Ho et al., 2022). Despite their success, U-Net models struggle with modeling long-range dependencies in complex, high-dimensional data. In response, the Diffusion Transformer (DiT) (Peebles & Xie, 2023; Chen et al., 2023; 2024a) is introduced, leveraging the inherent scalability of Transformers to efficiently enhance model capacities and handle more complex tasks with improved performance.

**Efficent diffusion.** Diverse methods have been proposed to tackle the poor real-time performance of diffusion models. These techniques fall into two main categories: reducing the number of sampling steps and decreasing the computational load per denoising step. In the first category, several works utilize distillation (Salimans & Ho, 2022; Luhman & Luhman, 2021) to obtain reduced sampling iterations. Furthermore, this category encompasses advanced techniques such as implicit samplers (Kong & Ping, 2021; Song et al., 2020a; Zhang et al., 2022) and specialized differential equation (DE) solvers. These solvers tackle both stochastic differential equations (SDE) (Song et al., 2020b; Jolicoeur-Martineau et al., 2021) and ordinary differential equations (ODE) (Lu et al., 2022a; Liu et al., 2022; Zhang & Chen, 2022), addressing diverse aspects of diffusion model optimization. In contrast, the second category mainly focuses on model compression. It leverages techniques like pruning (Fang et al., 2023; Zhang et al., 2024; Wang et al., 2024b) and quantization (Shang et al., 2023; Huang et al., 2024; He et al., 2024) to reduce the workload in a static way. Additionally, dynamic inference compression is also being explored (Liu et al., 2023; Pan et al., 2024), where different models are employed at varying steps of the process. In this work, we focus on the urgently needed DiT acceleration through feature cache, a method distinct from the above-discussed ones.

**Feature cache.** Due to the high similarity between activations (Li et al., 2023b; Wimbauer et al., 2024) across continuous denoising steps in diffusion models, recent studies (Ma et al., 2024c; Wimbauer et al., 2024; Li et al., 2023a) have explored caching these features for reuse in subsequent steps to avoid redundant computations. Notably, their strategies rely heavily on the specialized structure

of U-Net, *e.g.*, up-sampling blocks [2] or `SpatialTransformer` blocks [3]. Besides, FORA (Selvaraju et al., 2024) and $\Delta$-DiT (Chen et al., 2024b) further apply the feature cache mechanism to DiT. However, both methods select the cache position and lifespan in a handcrafted way. Learning-to-Cache (Ma et al., 2024b) introduces a learnable cache scheme but fails to harmonize training and inference. In this work, we design a new training framework, to alleviate the discrepancies between the training and inference, which further enhances the performance and acceleration ratio for DiT.

## 3 PRELIMILARIES

**Cache granularity.** The noise estimation network of DiT (Peebles & Xie, 2023) is built on the Transformer block (Vaswani, 2017), which is composed of an Attention block and a feed-forward network (FFN). Each Attention block and FFN is wrapped up in a residual connection (He et al., 2016). For convenience, we sequentially denote these Attention blocks and FFNs without residual connections as $\{b_0, b_1, \ldots b_{N-1}\}$, where $N$ is their total amount. Following Ma et al. (2024b), we store the output of $b_i$ in cache as $c_i$. The cache, once completely filled, is represented as follows:

$$\text{cache} = [c_0, c_1, \ldots, c_{N-1}]. \tag{1}$$

**Cache router.** The cache scheme for DiT can be formulated with a pre-defined threshold $\tau$ ($0 \leq \tau < 1$) and a customized router matrix:

$$\text{Router} = [r_{t,i}]_{1 \leq t \leq T, 0 \leq i \leq N-1} \in \mathbb{R}^{T \times N}, \tag{2}$$

where $0 < r_{t,i} \leq 1$ and $T$ is the maximum denoising step. At timestep $t$ during inference, the residual corresponding to $b_i$ is fused with $o_i$ defined as follows:

$$o_i = \begin{cases} b_i(\mathbf{h}_i, \mathbf{cs}), & r_{t,i} > \tau \\ c_i, & r_{t,i} \leq \tau \end{cases}, \tag{3}$$

where $\mathbf{h}_i$ is the image feature and $\mathbf{cs}$ represents the conditional inputs [4]. Specifically, $r_{t,i} > \tau$ indicates computing $b_i(\mathbf{h}_i, \mathbf{cs})$ as $o_i$. This computed output also replaces $c_i$ in the `cache`. Otherwise, the model loads $c_i$ from `cache` without computation. Here we present a naive example of the cache scheme as depicted in Fig. 2. To be noted, $\text{Router}_{T,:}$ is set to $[1]_{1 \times N}$ by default to pre-fill the empty `cache`.

**Cache usage ratio (CUR).** In addition, we define cache usage ratio (CUR) formulated as $\frac{\sum_{t=1}^{t=T} \sum_{i=0}^{N-1} \mathbb{I}_{r_{t,i} \leq \tau}}{N \times T}$ in this paper to represent the reduced computation by reusing cached features. For instance, CUR is roughly equal to 33.33% in Fig. 2.

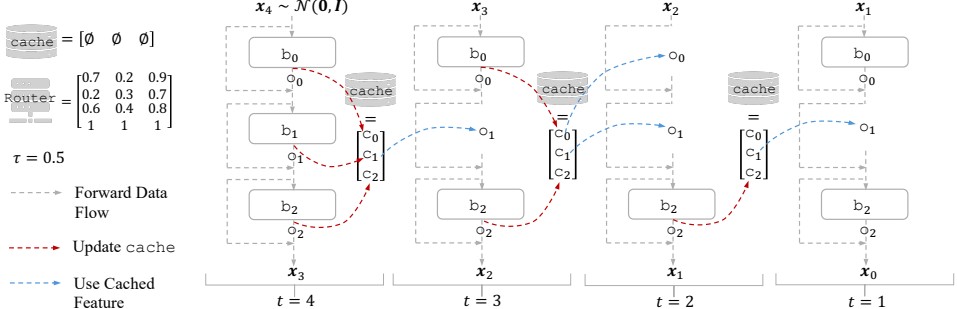

Figure 2: Generation process from a random Gaussian noise $\boldsymbol{x}_4$ to an image $\boldsymbol{x}_0$ using feature cache ($T = 4, N = 3$). We omit the sampler (Ho et al., 2020; Song et al., 2020a) and conditional inputs.

---

[2]`https://github.com/CompVis/stable-diffusion/blob/main/ldm/modules/diffusionmodules/openaimodel.py#L626`

[3]`https://github.com/CompVis/stable-diffusion/blob/main/ldm/modules/attention.py#L218`

[4]For example, $\mathbf{cs}$ represents the time condition and textual condition for text-to-image (T2I) generation.

## 4 HARMONICA

In this section, we first observe that the existing learning-based feature cache strategy shows discrepancies between the training and inference (Sec. 4.1). Then, we propose a framework named **HarmoniCa** to **harmoni**ze them for better feature **ca**che (Sec. 4.2). Finally, our HarmoniCa shows higher efficiency and better applicability than the previous training-based method (Sec. 4.3).

### 4.1 DISCREPANCY BETWEEN TRAINING AND INFERENCE

Revealing previous approaches for DiT, most of them (Selvaraju et al., 2024; Chen et al., 2024b) manually set the value of the `Router` in a heuristic way. To be adaptive, Learning-to-Cache (Ma et al., 2024b) employs a learnable `Router` [5]. However, we have identified two discrepancies between its training and inference phases in the following.

**Prior timestep disregard.** As illustrated in Fig. 2, the inference process employing feature cache at timestep $t$ is subject to the prior timesteps. For example, at timestep $t = 1$, the input $x_1$ has the error induced by reusing the cached features $c_0$ and $c_1$ at preceding timestep $t = 2$. Furthermore, reusing and updating features at earlier timesteps also shape the contents of the current `cache`.

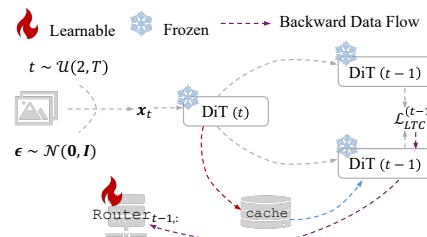

However, Learning-to-Cache is unaffected by prior denoising steps during training. Specifically, for each training iteration, as depicted in Fig. 3 (a), it first uniformly samples a timestep $t$ akin to DDPM (Ho et al., 2020). It then pre-fills an empty `cache` at $t$ and proceeds to train `Router`$_{t-1,:}$ at subsequent timestep $t - 1$, without being influenced by the feature cache mechanism from timestep $T$ to $t + 1$.

Figure 3: Training paradigm of Learning-to-Cache. $\mathcal{L}_{LTC}^{(t)}$ denotes the loss function. In each iteration, this method manually adds noise to images to obtain $x_t$ as the input of DiT at $t$. "$*$" in "DiT $(*)$" represents the current timestep.

**Objective mismatch.** Moreover, we also find that Learning-to-Cache (Ma et al., 2024b) solely focuses on the predicted noise at each denoising step during training. It leverages the following learning objective at timestep $t$:

$$\mathcal{L}_{LTC}^{(t)} = \mathcal{L}_{MSE}^{(t)} + \beta \sum_{i=0}^{N-1} r_{t,i}, \tag{4}$$

where $\beta$ is a coefficient for the regularization term of the `Router`$_{t:}$ and $\mathcal{L}_{MSE}^{(t)}$ represents the Mean Square Error (MSE) between the predicted noise of the DiT with and without reusing cached features at $t$.

In contrast, the target during inference is to generate the high-quality image $x_0$, which also leads to a discrepancy of objective.

### 4.2 HARMONIZING TRAINING AND INFERENCE

Existing studies (Ning et al., 2023; Li et al., 2024b; Ning et al., 2024) on diffusion models show that discrepancies between training and inference phases can lead to error accumulation (Arora et al., 2022; Schmidt, 2019) and results in performance degradation. Therefore, we **harmoni**ze training and inference with a new learning-based **ca**ching framework called **HarmoniCa**. It is composed of the following two techniques to alleviate the discrepancies mentioned above. Detailed algorithms of HarmoniCa can be found in Sec. A.

**Step-wise denoising training.** To mitigate the first discrepancy, as shown in Fig. 4 (a), we propose a new training paradigm named *Step-Wise Denoising Traning* (SDT), which completes the entire denoising process over $T$ timesteps, thereby accounting for the `cache` usage and update from all prior timesteps. Specifically, at timestep $T$, we randomly sample a Gaussian noise $x_T$ and perform a single denoising step to pre-fill the `cache`. Over the following $T - 1$ timesteps, the student model, which employs the feature cache mechanism, gradually removes noise to generate an image.

---

[5] $r_{t,i}$ in the `Router` is a learnable parameter.

Concurrently, the teacher model executes the same task without utilizing the `cache`. Requiring the student to mimic the output representation of its teacher, we compute the loss function and perform back-propagation to update $\text{Router}_{t,:}$ at each timestep $t$. To ensure that each $\text{r}_{t,i}$ is differentiable during training, distinct from Eq. (3), we proportionally combine the directly computed feature with the cached one to obtain $\text{o}_i$ following Ma et al. (2024b):

$$\text{o}_i = \text{r}_{t,i}\text{b}_i(\mathbf{h}_i, \mathbf{cs}) + (1 - \text{r}_{t,i})\text{c}_i. \tag{5}$$

Similar to inference, we also update $\text{c}_i$ in the `cache` with $\text{b}_i(\mathbf{h}_i, \mathbf{cs})$ when $\text{r}_{t,i} > \tau$. To improve training stability (Wimbauer et al., 2024), we fetch the output from the student as the input to the teacher for the next iteration. We repeat the above $T$ learning iterations until the end of training.

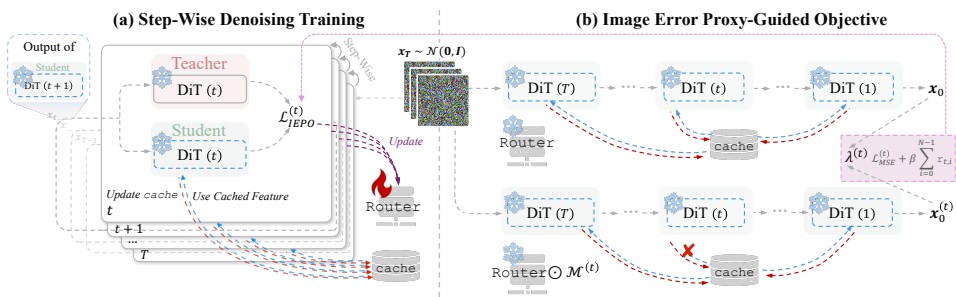

Figure 4: Overview of HarmoniCa. (a) *Step-Wise Denoising Training* (SDT) mimics the multi-timestep inference stage, which integrates the impact of prior timesteps at $t$. (b) *Image-Error Proxy-Guided Objective* (IEPO) incorporates the final image error into the learning objective by an efficient proxy $\lambda^{(t)}$, which is updated through gradient-free image generation passes every `C` training iterations. $\mathcal{M}^{(t)}$ masks the `Router` to disable the impact of the cache mechanism at $t$. $\odot$ denotes the operation of element-wise multiplication.

As depicted in Fig. 5, by incorporating prior denoising timesteps during training, SDT significantly reduces error at each timestep and obtains a much more accurate image $x_0$, even with lower computation, compared to Learning-to-Cache.

**Image error proxy-guided objective.** For the second discrepancy, a straightforward solution to align the target with inference involves using the error of final image $x_0$ caused by `cache` usage directly with a regularization term of `Router` as our training objective. However, even for DiT-XL/2 $256\times256$ (Peebles & Xie, 2023) with a small training batch size, this requires approximately $5\times$ GPU memory and $10\times$ time compared to SDT combined with $\mathcal{L}_{LTC}^{(t)}$ as detailed in Sec. B, making it impractical. Therefore, we have to identify a proxy for the error of $x_0$ that can be integrated into the learning objective.

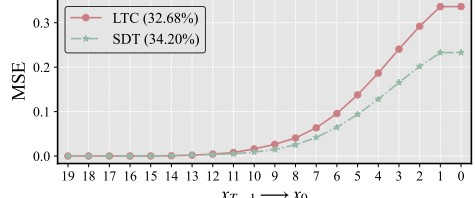

Figure 5: MSE of $x_t$ for DiT-XL/2 $256 \times 256$ (Peebles & Xie, 2023) ($T = 20, N = 56$) induced by different feature cache methods. $x_t$ is the noisy image obtained at timestep $t + 1$. "LTC" denotes Learning-to-Cache. For a fair comparison, $\mathcal{L}_{LTC}^{(t)}$ is employed for SDT. We mark the CUR in the brackets.

Based on the above analysis, we propose an *Image Error Proxy-guided Objective* (IEPO). It is defined at each timestep $t$ as follows:

$$\mathcal{L}_{IEPO}^{(t)} = \lambda^{(t)}\mathcal{L}_{MSE}^{(t)} + \beta \sum_{i=0}^{N-1} \text{r}_{t,i}, \tag{6}$$

where $\lambda^{(t)}$ is our final image error proxy treated as a coefficient of $\mathcal{L}_{MSE}^{(t)}$. This proxy represents the final image error caused by the `cache` usage at $t$. With a large $\lambda^{(t)}$, $\mathcal{L}_{MSE}^{(t)}$ prioritizes reduction of the output error at $t$. This tends to decrease the cached feature usage rate at the corresponding timestep, and vice versa. Therefore, our proposed objective considers the trade-off between the error of $x_0$ and the `cache` usage at a certain denoising step.

Here, we detail the process to obtain $\lambda^{(t)}$ as follows. For a given `Router`, a mask matrix is defined to disable the use of cached features and force updating the entire `cache` at $t$ as:

$$\mathcal{M}_{j,k}^{(t)} = \begin{cases} 1, & j \neq t \\ \frac{1}{\text{r}_{j,k}}, & j = t \end{cases}, \tag{7}$$

where $(j, k)$ [6] denotes the index of $\mathcal{M}^{(t)} \in \mathbb{R}^{T \times N}$. As depicted in Fig. 4 (b), $\boldsymbol{x}_0$ and $\boldsymbol{x}_0^{(t)}$ are final images generated from a randomly sampled Gaussian noise $\boldsymbol{x}_T$ using feature cache guided by (Upper) `Router` and (Lower) `Router` element-wise multiplied by $\mathcal{M}^{(t)}$, respectively. Then, we can formulate $\lambda^{(t)}$ as:

$$\lambda^{(t)} = \|\boldsymbol{x}_0 - \boldsymbol{x}_0^{(t)}\|_F^2, \tag{8}$$

where $\|\cdot\|_F$ denotes the Frobenius norm. To adapt to the training dynamics, we periodically update all the coefficients $\{\lambda^{(1)}, \ldots, \lambda^{(T)}\}$ every `C` iterations [7], instead of employing static ones.

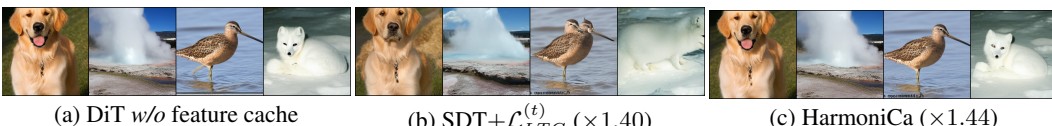

(a) DiT *w/o* feature cache     (b) SDT+$\mathcal{L}_{LTC}^{(t)}$ ($\times 1.40$)     (c) HarmoniCa ($\times 1.44$)

Figure 6: Random samples for DiT-XL/2 $256 \times 256$ (Peebles & Xie, 2023) *w/* and *w/o* feature cache ($T = 20$). We mark the speedup ratio in the brackets.

Fig. 6 shows that $\mathcal{L}_{IEPO}^{(t)}$ helps yield much more accurate objective-level traits and significantly improves the quality of $\boldsymbol{x}_0$ even at a higher speedup ratio than $\mathcal{L}_{LTC}^{(t)}$. The study in Sec. C justifies that employing $\mathcal{L}_{LTC}^{(t)}$ incurs the optimization deviating from minimizing the error of $\boldsymbol{x}_0$.

### 4.3 EFFICIENCY DISCUSSION

**Training efficiency.** Our HarmoniCa incurs significantly lower training costs than the previous learning-based method. As shown in Tab. 1, HarmoniCa requires no training images, whereas Learning-to-Cache utilizes original training datasets. Thus, it is challenging to apply Learning-to-Cache to models like the PIXART-$\alpha$ (Chen et al., 2023) family, which are trained on large datasets, limiting its applicability. Moreover, while dynamic update of $\lambda^{(t)}$ incurs approximately 10% extra time overhead, HarmoniCa requires only three-quarters of the training hours compared to Learning-to-Cache, which needs to pre-fill the `cache` for each training iteration.

**Inference efficiency.** Fortunately, our method with a pre-learned `Router` has no computational overhead during runtime. Moreover, less than 6% extra memory overhead [8] is induced by `cache` for DiT-XL/2 $256 \times 256$ with a batch size of 8. Therefore, the introduced inference cost is controlled at a small level.

Table 1: Training costs of learning-based feature cache methods for DiT-XL/2 $256 \times 256$ (Peebles & Xie, 2023) ($T = 20$). We train with all methods for 20K iterations using a global batch size 64 on 4 NVIDIA H800 80GB GPUs. For HarmoniCa, we set `C` = 500. As in the original paper, we utilize the full ImageNet training set (Russakovsky et al., 2015) for Learning-to-Cache.

| Method | #Images | Time(h) | Memory(GB/GPU) |
|---|---|---|---|
| Learning-to-Cache | 1.22M | 2.15 | 33.33 |
| SDT+$\mathcal{L}_{LTC}^{(t)}$ | 0 | 1.47 | 33.28 |
| HarmoniCa | 0 | 1.63 | 33.28 |

## 5 EXPERIMENTS

This section begins by outlining the detailed experimental protocols (Sec. 5.1). Subsequently, we provide comprehensive comparisons across different methods to show the superior performance and acceleration ratio of our HarmoniCa (Sec. 5.2). Finally, we provide ablation studies for the key designs of our method (Sec. 5.3).

### 5.1 IMPLEMENTATION DETAILS

**Models and datasets.** We conduct experiments on two different image generation tasks. For class-conditional task, we employ DiT-XL/2 (Peebles & Xie, 2023) $256 \times 256$ and $512 \times 512$ models pre-trained and accessed on ImageNet dataset (Russakovsky et al., 2015). For text-to-image (T2I) task, we utilize PIXART-$\alpha$ (Chen et al., 2023) series, known for its outstanding performance. These

---

[6] $1 \leq j \leq T$ and $0 \leq k \leq N - 1$.

[7] `C` mod $T = 0$.

[8] The `cache` occupies 0.49 GB GPU memory and inference without the feature cache mechanism takes 8.18 GB GPU memory.

models including PIXART-XL/2 at resolutions of $256 \times 256$ and $512 \times 512$, along with PIXART-XL/2-1024-MS at a higher resolution of $1024 \times 1024$, are tested on MS-COCO dataset (Lin et al., 2015). We additionally use T5 model (Raffel et al., 2023) as their text encoders.

**Training settings.** Following Ma et al. (2024b), we set the threshold $\tau$ as 0.1 for all the models. Each of them is trained for 20K iterations employing the AdamW optimizer (Loshchilov & Hutter, 2019) on 4 NVIDIA H800 80GB GPUs. The learning rate is fixed at 0.01, $C$ is set to 500, and global batch sizes of 64, 48, and 32 are utilized for models with increasing resolutions. Additionally, we collect 1000 MS-COCO captions for T2I training.

**Baselines.** For class-conditional experiments, we choose the current state-of-the-art (SOTA) Learning-to-Cache (Ma et al., 2024b) as our baseline. Due to the limits mentioned in Sec. 4.3, we employ FORA (Selvaraju et al., 2024) and $\Delta$-DiT (Chen et al., 2024b), excluding Learning-to-Cache for the T2I task. The results of these methods are obtained either by re-running their open-source code (if available) or by using the data provided in the original papers, all under the same conditions as our experiments. We also report the performance of models with reduced denoising steps.

**Evaluation.** To assess the generation quality, Fréchet Inception Distance (FID) (Nash et al., 2021), and sFID (Nash et al., 2021) are applied to all experiments. For DiT/XL-2, we additionally provide Inception Score (IS) (Salimans et al., 2016), Precision, and Recall (Kynkäänniemi et al., 2019) as reference metrics. For PIXART-$\alpha$, to gauge the compatibility of image-caption pairs, we calculate CLIP score (Hessel et al., 2022) using ViT-B/32 (Dosovitskiy et al., 2020) as the backbone. To evaluate the inference efficiency, we measure the CUR [9] and the inference latency for a batch size of 8. In detail, we sample 50K images adopting DDIM (Song et al., 2020a) for DiT-XL/2, and 30K images utilizing IDDPM (Nichol & Dhariwal, 2021), DPM-Solver++ (Lu et al., 2022b), and SA-Solver (Xue et al., 2024) for PIXART-$\alpha$. All of them use classifier-free guidance (cfg) (Ho & Salimans, 2022).

More implementation details can be found in Sec. D and the results of PIXART-$\Sigma$ (Chen et al., 2024a) family are available in Sec. E, including generation with an extremely high-resolution of $2048 \times 2048$. In addition, we also present the results of combination with quantization to further accelerate DiT inference in Sec. F.

## 5.2 MAIN RESULTS

Table 2: Accelerating image generation on ImageNet for the DiT-XL/2. We mark the speedup ratio in the brackets and highlight the **best score** in bold.

| Method | T | IS↑ | FID↓ | sFID↓ | Prec.↑ | Recall↑ | CUR(%)↑ | Latency(s)↓ |
|---|---|---|---|---|---|---|---|---|
| DiT-XL/2 $256 \times 256$ (cfg = 1.5) | | | | | | | | |
| DDIM (Song et al., 2020a) | 50 | 240.37 | 2.27 | 4.25 | 80.25 | 59.77 | - | 1.767 |
| DDIM (Song et al., 2020a) | 39 | 237.84 | 2.37 | 4.32 | 80.22 | 59.31 | - | $1.379_{(\times 1.28)}$ |
| Learning-to-Cache (Ma et al., 2024b) | 50 | 233.26 | 2.62 | 4.50 | 79.40 | 59.15 | 23.39 | $1.419_{(\times 1.25)}$ |
| HarmoniCa | 50 | **238.74** | **2.36** | **4.24** | **80.57** | **59.68** | **23.68** | **1.361**$_{(\times 1.30)}$ |
| DDIM (Song et al., 2020a) | 20 | 224.37 | 3.52 | 4.96 | 78.47 | 58.33 | - | 0.658 |
| DDIM (Song et al., 2020a) | 14 | 201.83 | 5.77 | 6.61 | 75.14 | 55.08 | - | $0.466_{(\times 1.41)}$ |
| Learning-to-Cache (Ma et al., 2024b) | 20 | 201.37 | 5.34 | 6.36 | 75.04 | 56.09 | 35.60 | $0.468_{(\times 1.41)}$ |
| HarmoniCa | 20 | **206.57** | **4.88** | **5.91** | **75.20** | **58.74** | **37.50** | **0.456**$_{(\times 1.44)}$ |
| DDIM (Song et al., 2020a) | 10 | 159.93 | 12.16 | 11.31 | 67.10 | 52.27 | - | 0.332 |
| DDIM (Song et al., 2020a) | 9 | 140.37 | 16.54 | 14.44 | 62.63 | 50.08 | - | $0.299_{(\times 1.11)}$ |
| Learning-to-Cache (Ma et al., 2024b) | 10 | 145.09 | 14.59 | 11.58 | 64.03 | 52.06 | 19.11 | $0.279_{(\times 1.19)}$ |
| HarmoniCa | 10 | **151.83** | **13.35** | **11.13** | **65.22** | **52.18** | **22.86** | **0.270**$_{(\times 1.23)}$ |
| DiT-XL/2 $512 \times 512$ (cfg = 1.5) | | | | | | | | |
| DDIM (Song et al., 2020a) | 20 | 184.47 | 5.10 | 5.79 | 81.77 | 54.50 | - | 3.356 |
| DDIM (Song et al., 2020a) | 16 | 173.31 | 6.47 | 6.67 | 81.10 | 51.30 | - | $2.688_{(\times 1.25)}$ |
| Learning-to-Cache (Ma et al., 2024b) | 20 | 178.11 | 6.24 | 7.01 | 81.21 | 53.30 | 23.57 | $2.633_{(\times 1.28)}$ |
| HarmoniCa | 20 | **179.84** | **5.72** | **6.61** | **81.33** | **55.80** | **25.98** | **2.574**$_{(\times 1.30)}$ |

**Class-conditional generation.** We begin our evaluation with DiT-XL/2 on ImageNet and compare it with current SOTA Learning-to-Cache (Ma et al., 2024b) and the approach employing fewer

---

[9]Definition can be found in Sec. 3.

timesteps. The results are presented in Tab. 2, where our HarmoniCa surpasses baseline methods. Notably, with a higher speedup ratio for a 10-step DiT-XL/2 $256 \times 256$, HarmoniCa achieves an FID of 13.35 and an IS of 151.83, outperforming Learning-to-Cache by 1.24 and 6.74, respectively. Moreover, the superiority of our HarmoniCa increases as the number of timesteps decreases. We conjecture that it is because the difficulty to learn a `Router` rises as the timestep goes up. Additionally, we further conduct experiments with a lower CUR for this task in Sec. H.

**T2I generation.** We also present PixArt-$\alpha$ results in Tab. 3, comparing our HarmoniCa against FORA (Selvaraju et al., 2024) and the method using fewer timesteps. HarmoniCa outperforms these benchmarks across all metrics. For example, with the 20-step DPM-Solver++, PIXART-$\alpha$ $256 \times 256$ employing HarmoniCa achieves an FID of 27.61 and speeds up by $1.52\times$, surpassing the non-accelerated model's FID of 27.68. In contrast, DPM-Solver++ with 15 steps and FORA only achieves FIDs of 31.68 and 38.20, respectively, with speed increases under $1.32\times$. Notably, HarmoniCa also cuts about 36% off processing time without dropping performance when using the IDDPM sampler, while FORA results in over a 20 FID increase and a 15.67% CUR decrease. Overall, our method consistently delivers superior performance and speedup improvements across different resolutions and samplers, demonstrating its efficacy. HarmoniCa also significantly outperforms $\Delta$-DiT (Chen et al., 2024b), which can be found in Sec. I.

Table 3: Accelerating image generation on MS-COCO for the PIXART-$\alpha$.

| Method | T | CLIP↑ | FID↓ | sFID↓ | CUR(%)↑ | Latency(s)↓ |
|---|---|---|---|---|---|---|
| PIXART-$\alpha$ $256 \times 256$ (cfg = 4.5) | | | | | | |
| DPM-Solver++ (Lu et al., 2022b) | 20 | 30.96 | 27.68 | 36.39 | - | 0.553 |
| DPM-Solver++ (Lu et al., 2022b) | 15 | 30.77 | 31.68 | 38.92 | - | 0.418$_{(\times 1.32)}$ |
| FORA (Selvaraju et al., 2024) | 20 | - | 38.20 | - | 50.00 | 0.424$_{(\times 1.30)}$ |
| HarmoniCa | 20 | **30.93** | **27.61** | **37.48** | 65.02 | **0.364**$_{(\times 1.52)}$ |
| IDDPM (Nichol & Dhariwal, 2021) | 100 | 31.25 | 24.15 | 33.65 | - | 2.572 |
| IDDPM (Nichol & Dhariwal, 2021) | 75 | 31.25 | 24.17 | 33.73 | - | 1.868$_{(\times 1.37)}$ |
| FORA (Selvaraju et al., 2024) | 100 | - | 55.30 | - | 50.00 | 1.889$_{(\times 1.36)}$ |
| HarmoniCa | 100 | **31.23** | **23.79** | **32.49** | 65.67 | **1.641**$_{(\times 1.56)}$ |
| SA-Solver (Xue et al., 2024) | 25 | 31.31 | 23.76 | 34.93 | - | 0.891 |
| SA-Solver (Xue et al., 2024) | 20 | 31.28 | 23.96 | 35.63 | - | 0.677$_{(\times 1.31)}$ |
| HarmoniCa | 25 | 31.29 | 23.85 | 35.56 | 54.31 | **0.665**$_{(\times 1.34)}$ |
| PIXART-$\alpha$ $512 \times 512$ (cfg = 4.5) | | | | | | |
| DPM-Solver++ (Lu et al., 2022b) | 20 | 31.30 | 23.96 | 40.34 | - | 1.759 |
| DPM-Solver++ (Lu et al., 2022b) | 15 | 31.29 | 25.12 | 40.37 | - | 1.291$_{(\times 1.36)}$ |
| HarmoniCa | 20 | **31.30** | 24.99 | 40.36 | 55.01 | **1.168**$_{(\times 1.51)}$ |
| PIXART-$\alpha$ $1024 \times 1024$ (cfg = 4.5) | | | | | | |
| DPM-Solver++ (Lu et al., 2022b) | 20 | 31.10 | 25.01 | 37.80 | - | 9.470 |
| DPM-Solver++ (Lu et al., 2022b) | 15 | 31.07 | 25.77 | 42.50 | - | 7.141$_{(\times 1.32)}$ |
| HarmoniCa | 20 | 31.08 | **24.76** | **41.83** | 59.65 | **6.289** $_{(\times 1.51)}$ |

## 5.3 ABLATION STUDIES

In this subsection, we employ a 20-step DDIM (Song et al., 2020a) sampler for DiT-XL/2 $256 \times 256$ and settings in Sec. 5.1 without special claim.

Table 4: Ablation results of different components. The first row denotes the model *w/o* feature cache. The second and last rows denote Learning-to-Cache and HarmoniCa, respectively.

| Training Paradigm | | Learning Objective | | IS↑ | FID↓ | sFID↓ | CUR(%)↑ | Latency(s)↓ |
|---|---|---|---|---|---|---|---|---|
| Learning-to-Cache | SDT | $\mathcal{L}_{LTC}^{(t)}$ | $\mathcal{L}_{IEPO}^{(t)}$ | | | | | |
| | | | | 224.37 | 3.52 | 4.96 | - | 0.658 |
| ✔ | | ✔ | | 115.00 | 18.57 | 16.18 | 32.68 | 0.483$_{(\times 1.36)}$ |
| ✔ | | | ✔ | 203.41 | 5.20 | 6.07 | 36.70 | 0.458$_{(\times 1.44)}$ |
| | ✔ | ✔ | | 166.65 | 8.01 | 7.62 | 34.20 | 0.471$_{(\times 1.40)}$ |
| | ✔ | | ✔ | **206.67** | **4.88** | **5.91** | 37.50 | **0.456**$_{(\times 1.44)}$ |

**Effect of different components.** To show the effectiveness of components involved in HarmoniCa, we apply different combinations of training techniques and show the results in Tab. 4. For the training paradigm, equipped with $\mathcal{L}_{LTC}^{(t)}$, our SDT significantly decreases FID by 10 compared to that of Learning-to-Cache. For the learning objective, our IEPO achieves nearly a 40 IS improvement and a

3.13 FID reduction for SDT compared with $\mathcal{L}_{LTC}^{(t)}$. Moreover, both SDT and IEPO can help significantly enhance performance for the counterparts in the table. For a fair comparison, we modify the implementation of Learning-to-Cache to train the entire `Router` in Tab. 4. A detailed discussion of this can be found in Sec. J.

**Effect of iteration interval** `C`**.** As illustrated in Fig. 7, we carry out experiments to evaluate the impact of varying `C` values on updating $\lambda^{(t)}$ in Eq. (8). Despite similar speedup ratios, using an extreme `C` value leads to notable performance degradation. Specifically, a large `C` means the proxy $\lambda^{(t)}$ fails to accurately and timely reflect the cache mechanism's effect on the final image. Conversely, a small `C` results in overly frequent updates, complicating train-

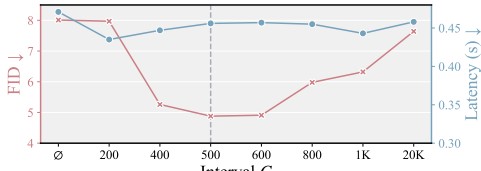

Figure 7: Ablation results of iteration interval `C`. $\varnothing$ denotes the model employing $\mathcal{L}_{LTC}^{(t)}$ as its loss function.

ing convergence. Hence, we choose a moderate value of 500 as `C` in this paper based on its superior performance, as demonstrated in the figure.

**Effect of coefficient** $\beta$**.** We also explore the trade-off between inference speed and performance for different values of $\beta$ in Eq. (6). As shown in Fig. 8, a higher $\beta$ leads to greater acceleration but at the cost of more pronounced performance degradation, and vice versa. Notably, performance declines gradually when $\beta \leq 8e^{-8}$ and more sharply outside this range. This observation suggests the potential for autonomously finding an optimal $\beta$ to balance speed and performance, which we aim to address in future research.

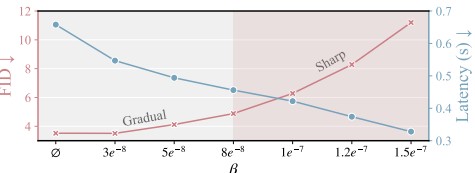

Figure 8: Ablation results of coefficient $\beta$ in Eq. (6). $\varnothing$ denotes the model *w/o* feature cache.

**Effect of different metrics for** $\lambda^{(t)}$**.** In Tab. 5, we conduct experiments to explore the effect of $\lambda^{(t)}$ with different metrics. Both $\|\cdot\|_F^2$ and $\mathcal{D}_{KL}(\cdot)$ lead to notable performance enhancements compared to using only the output error (*i.e.*, $\lambda^{(t)} = 1$) at each time step. Due to the insensitivity to outliers, $\sum |\cdot|$ is generally less effective for image reconstruction and inferior to the others in Tab. 5.

Table 5: Ablation results of different metrics for $\lambda^{(t)}$. The first and second columns represent the model *w/o* feature cache and SDT+$\mathcal{L}_{LTC}^{(t)}$, respectively. $\mathcal{D}_{KL}(\cdot)$ denotes Kullback–Leibler (KL) divergence.

| $\lambda^{(t)}$ | $+\infty$ | 1 | $\sum \lvert \boldsymbol{x}_0 - \boldsymbol{x}_0^{(t)} \rvert$ | $\lVert \boldsymbol{x}_0 - \boldsymbol{x}_0^{(t)} \rVert_F^2$ | $\mathcal{D}_{KL}(\boldsymbol{x}_0, \boldsymbol{x}_0^{(t)})$ |
|---|---|---|---|---|---|
| IS↑ | 224.37 | 166.65 | 172.08 | **206.57** | 205.91 |
| FID↓ | 3.52 | 8.01 | 6.95 | **4.88** | 5.25 |
| sFID↓ | 4.96 | 7.62 | 7.79 | 5.91 | **5.51** |
| CUR(%)↑ | - | 34.20 | 34.82 | **37.50** | 36.79 |
| Latency(s)↓ | 0.658 | 0.471$_{(\times 1.40)}$ | 0.470$_{(\times 1.40)}$ | **0.456**$_{(\times 1.44)}$ | 0.458$_{(\times 1.44)}$ |

# 6 CONCLUSION

In this research, we focus on accelerating Diffusion Transformers (DiTs) through the cache mechanism in a learning-based way. We first identify two discrepancies between training and inference of the previous method: (1) *Prior Timestep Disregard* in which earlier step influences are neglected, leading to inconsistency with inference, and (2) *Objective Mismatch*, where training focuses on intermediate results, misaligning with the final image quality target. To alleviate these discrepancies, we **harmoni**ze training and inference by introducing a novel feature **ca**che framework dubbed **HarmoniCa**, which consists of the *Step-wise Denoising Training* (SDT) and the *Image Error-Aware Optimization Objective* (IEPO). SDT captures the influence of all timesteps during training, closing the gap with the inference stage, while IEPO introduces an efficient proxy for final image error, ensuring that optimization objectives remain aligned with inference requirements. With the combination of the two components, extensive experiments demonstrate that our framework achieves superior performance and efficiency with significantly lower training cost compared to the existing training-based method.

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

# Appendix

We organize the appendix as follows.

- In Sec. A, we provide the detailed procedure of HarmoniCa.
- In Sec. B, we analyze why directly employing the final image error with a regularization term as the loss function is not feasible.
- In Sec. C, we investigate the optimization deviation of overlooking the final image error during training.
- In Sec. D, we introduce more details about implementation and other hyper-parameters.
- In Sec. E, we adapt HarmoniCa to PIXART-$\Sigma$ and show the promising performance.
- In Sec. F, we combine the quantization with HarmoniCa to show further acceleration.
- In Sec. G, we introduce the implementation details of model quantization employed in Sec. F.
- In Sec. H, we compare HarmoniCa with Learning-to-Cache under a relatively low CUR(%).
- In Sec. I, we compare HarmoniCa with $\Delta$-DiT.
- In Sec. J, we compare HarmoniCa with Learning-to-Cache with different sampling strategies.
- In Sec. K, we conduct experiments comparing HarmoniCa with additional caching-based acceleration methods.
- In Sec. L, we compare HarmoniCa with quantization and pruning methods.
- In Sec. M, we conduct more experiments on different metrics for image error proxy $\lambda^{(t)}$.
- In Sec. N, we study the effect of applying the trained `Router` to a different sampler.
- In Sec. O, we compare HarmoniCa with Learning-to-Cache as the speedup ratio increases.
- In Sec. P, we conduct more experiments with SA-Solver under different configurations to show the effectiveness of HarmoniCa.
- In Sec. Q, we show the remarkable performance and acceleration ratio achieved by HarmoniCa on more high-quality datasets with additional metrics.
- In Sec. R, we provide ablation results of HarmoniCa across different thresholds $\tau$.
- In Sec. S, we show quantitative comparison (Fig. C and D) with some analysis.
- In Sec. T, we show more visualization results (Fig. E to K) across different model series and resolutions.

# A ALOGRITHM OF HARMONICA

As described in Alg. 1, we provide a detailed algorithm of our HarmoniCa. For clarity, we omit the pre-fill stage (*i.e.*, denoising at $T$), where $\texttt{Router}_{T:}$ is forced to be set to $\{1\}_{1 \times N}$. The $\texttt{conds}$ for T2I tasks and class-conditional generation are pre-prepared text prompts and class labels, respectively.

---

**Algorithm 1** HarmoniCa: the upper snippet describes the full procedure, and the lower side contains the subroutine for computing the proxy of the final image error.

---

func HARMONICA($\phi, \boldsymbol{\epsilon}_\theta, \texttt{iters}, \texttt{conds}, \tau, \beta, T, \texttt{C}$)

**Require:** $\phi(\cdot)$ — diffusion sampler
$\qquad\quad \boldsymbol{\epsilon}_\theta(\cdot)$ — DiT model
$\qquad\quad \texttt{iters}$ — amount of training iterations
$\qquad\quad \texttt{conds}$ — conditional inputs
$\qquad\quad \tau$ — threshold
$\qquad\quad \beta$ — constraint coefficient
$\qquad\quad T$ — maximum denoising step
$\qquad\quad \texttt{C}$ — iteration interval

1: Initialize $\texttt{Router}$ with a normal distribution
2: $\texttt{cache} = \emptyset$ $\qquad\qquad\qquad\qquad\qquad\qquad\qquad\qquad\qquad\qquad$ ▷ Initialize $\texttt{cache}$
3: **for** $i$ in 0 to $\frac{\texttt{iters}}{T} - 1$ **do**:
4: $\quad \boldsymbol{x}_T \sim \mathcal{N}(\mathbf{0}, \mathbf{I})$
5: $\quad$ **if** $i\%\frac{\texttt{C}}{T} = 0$ **then**
6: $\quad\quad \{\lambda^{(1)}, \dots, \lambda^{(T)}\} = \texttt{gen\_proxy}(\phi, \boldsymbol{\epsilon}_\theta, \boldsymbol{x}_T, \texttt{conds}[i], \tau, \texttt{Router})$
7: $\quad$ **end if**
8: $\quad$ **for** $t$ in $T$ to 1 **do**:
9: $\quad\quad \boldsymbol{\epsilon}^{(t)'} = \boldsymbol{\epsilon}_\theta(\boldsymbol{x}_t, t, \texttt{conds}[i], \texttt{Router}_{t,:}, \tau, \texttt{cache})$ $\qquad\qquad$ ▷ Fig. 2
10: $\quad\quad \boldsymbol{\epsilon}^{(t)} = \boldsymbol{\epsilon}_\theta(\boldsymbol{x}_t, t, \texttt{conds}[i])$
11: $\quad\quad \mathcal{L}_{IEPO}^{(t)} = \lambda^{(t)}\|\boldsymbol{\epsilon}^{(t)'} - \boldsymbol{\epsilon}^{(t)}\|_F^2 + \beta\sum_{i=0}^{N-1} \texttt{r}_i^{(t)}$ $\qquad\qquad$ ▷ Eq. (6)
12: $\quad\quad$ Tune $\texttt{Router}_{t,:}$ by back-propagation
13: $\quad\quad \boldsymbol{x}_{t-1} = \phi(\boldsymbol{x}_t, t, \boldsymbol{\epsilon}^{(t)'})$
14: $\quad$ **end for**
15: **end for**
16: **return** $\texttt{Router}$

---

func gen_proxy($\phi, \boldsymbol{\epsilon}_\theta, \boldsymbol{x}_T, \texttt{cond}, \tau, \texttt{Router}$)

1: $\texttt{cache} = \emptyset$ $\qquad\qquad\qquad\qquad\qquad\qquad\qquad\qquad\qquad\qquad$ ▷ Initialize $\texttt{cache}$
2: Employ feature cache guided by $\texttt{Router}$ to generate $\boldsymbol{x}_0$
3: **for** $t$ in $T$ to 1 **do**:
4: $\quad$ Generate $\mathcal{M}^{(t)}$ $\qquad\qquad\qquad\qquad\qquad\qquad\qquad\qquad\qquad$ ▷ Eq. (7)
5: $\quad$ Employ feature cache guided by $\texttt{Router} \odot \mathcal{M}^{(t)}$ to generate $\boldsymbol{x}_0^{(t)}$
6: $\quad \lambda^{(t)} = \|\boldsymbol{x}_0 - \boldsymbol{x}_0^{(t)}\|_F^2$ $\qquad\qquad\qquad\qquad\qquad\qquad\qquad$ ▷ Eq. (8)
7: **end for**
8: **return** $\{\lambda^{(1)}, \lambda^{(2)}, \dots, \lambda^{(T)}\}$

---

# B IMAGE ERROR WITH ROUTER REGULARIZATION TERM AS TRAINING OBJECTIVE

In Tab. A, SDT+$\mathcal{L}_{\boldsymbol{x}_0}^{(t)}$ requires $t-1$ additional denoising passes per training iteration at $t$ to compute the error of $\boldsymbol{x}_0$. Consequently, this approach consumes about $\times 9.73$ GPU hours compared to SDT+$\mathcal{L}_{LTC}^{(t)}$. Due to the extensive intermediate activations stored from timestep $t$ to 1 for back-propagation, it also costs $\times 4.90$ GPU memory. This estimation is conducted with small batch sizes and limited iterations. Therefore, SDT+$\mathcal{L}_{\boldsymbol{x}_0}^{(t)}$ is less feasible for models with larger latent spaces or higher token counts per image, such as DiT-XL/2 $512 \times 512$, particularly in large-batch, complete training scenarios. Additionally, the network effectively becomes $T \times N$ stacked Transformer blocks under this strategy, making it difficult (Wang et al., 2024a) to optimize the $\texttt{Router}$ with even a moderate $T$ value, such as 50 or 100.

Table A: Training costs estimation across different methods for DiT-XL/2 $256 \times 256$ (Peebles & Xie, 2023) ($T = 20$). We only employ 5K iterations with a global batch size of 8 on 4 NVIDIA H800 80G GPUs. $\mathcal{L}_{\boldsymbol{x}_0}^{(t)}$ denotes the loss function replacing $\mathcal{L}_{MSE}^{(t)}$ in Eq. (4) with the final image error.

| Method | #Images | Time(h) | Memory(GB/GPU) |
|---|---|---|---|
| SDT+$\mathcal{L}_{\boldsymbol{x}_0}^{(t)}$ | 0 | 1.46 | 65.36 |
| SDT+$\mathcal{L}_{LTC}^{(t)}$ | 0 | 0.15 | 13.33 |

## C  OPTIMIZATION DEVIATION

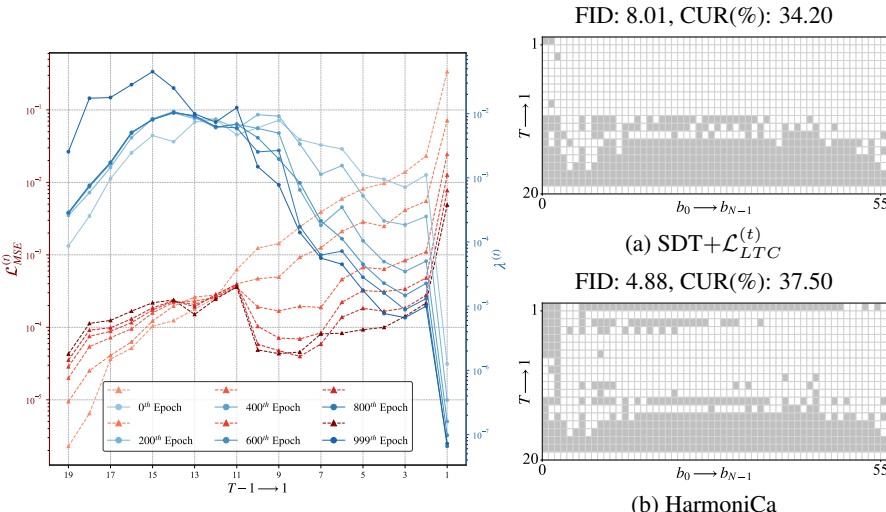

(a) SDT+$\mathcal{L}_{LTC}^{(t)}$

(b) HarmoniCa

Figure A: (Left) Variations of $\mathcal{L}_{MSE}^{(t)}$ and $\lambda^{(t)}$ for SDT+$\mathcal{L}_{LTC}^{(t)}$. (Right) Router visualization across different methods. The gray grid $(t, i)$ represents using the feature in cache at $t$ without computing $\mathsf{o}_i$. The white grid indicates computing and updating cache. We also mark their FID (Heusel et al., 2018) and CUR. All the above experiments employ DiT-XL/2 $256 \times 256$ ($T = 20, N = 56$).

To generate high-quality $\boldsymbol{x}_0$ and accelerate the inference phase, we believe only considering the output error at a certain timestep can cause a deviated optimization due to its gap *w.r.t* the error of $\boldsymbol{x}_0$. To validate this, we plot the values of $\mathcal{L}_{MSE}^{(t)}$ in Eq. (4) and $\lambda^{(t)}$ in Eq. (8) during the training phase of SDT+$\mathcal{L}_{LTC}^{(t)}$ in Fig. A (Left). Comparing $\mathcal{L}_{MSE}^{(t)}$ and $\lambda^{(t)}$ across different denoising steps, their results present a significant discrepancy. For instance, $\mathcal{L}_{MSE}^{(t)}$ at $t = 14$ is several orders of magnitude smaller than that at $t = 1$ during the entire training process, and the opposite situation happens for $\lambda^{(t)}$. Intuitively, this indicates that we could increase the cache usage rate at $t = 1$, and vice versa at $t = 14$ for higher performance while keeping the same speedup ratio according to the value of the proxy $\lambda^{(t)}$. However, only considering the output error at each timestep (*i.e.*, $\mathcal{L}_{MSE}^{(t)}$) can optimize towards a shifted direction. In practice, the learned Router with the guidance of $\lambda^{(t)}$ in Fig. A (Right) (b) caches less in large timesteps like $t = 14$ and reuses more in small timesteps as $t = 1$ compared to that in Fig. A (Right) (a) achieving significant performance enhancement.

## D  MORE IMPLEMENTATION DETAILS

In this section, we present more details on the implementation of our HarmoniCa. First, following Ma et al. (2024b), we also perform a sigmoid function [10] to each $\mathsf{r}_{t,i}$ before it is passed to the model. Moreover, unless specified otherwise, the hyper-parameter $\beta$ in Eq. (6) for all experiments is given in Tab. B; any exceptions are noted in the relevant tables.

---

[10]$\sigma(x) = \frac{1}{1+e^{-x}}$

Table B: Hyper-parameter $\beta$ for training the `Router`.

| Model | DiT-XL/2 | | | | PIXART-$\alpha$ | | | | | PIXART-$\Sigma$ | | |
|---|---|---|---|---|---|---|---|---|---|---|---|---|
| Resolution | $256 \times 256$ | | | $512 \times 512$ | $256 \times 256$ | | | $512 \times 512$ | $1024 \times 1024$ | $512 \times 512$ | $1024 \times 1024$ | $2048 \times 2048$ |
| $T$ | 10 | 20 | 50 | 20 | 20 | 100 | 25 | 20 | 20 | 20 | 20 | 20 |
| $\beta$ | $7e^{-8}$ | $8e^{-8}$ | $5e^{-8}$ | $4e^{-8}$ | $1e^{-3}$ | $8e^{-4}$ | $8e^{-4}$ | $8e^{-4}$ | $8e^{-4}$ | $1e^{-3}$ | $8e^{-4}$ | $8e^{-4}$ |

## E RESULTS FOR PIXART-$\Sigma$

In this section, we present the results for the PIXART-$\Sigma$ family, including PIXART-$\Sigma$-XL/2-512-MS, PIXART-$\Sigma$-XL/2-1024-MS, and PIXART-$\Sigma$-XL/2-2K-MS. For the latter one, we test by sampling 10K images. Additionally, we train the `Router` with a batch size of 16 and measure latency using a batch size of 1. All other settings are consistent with those described in Sec. 5.1.

As shown in Table C, HarmoniCa achieves a $\times 1.51$ speedup along with improved CLP scores and sFID compared to the non-accelerated model for PIXART-$\Sigma$ $2048 \times 2048$. Notably, this is the first time for the feature cache mechanism to accelerate image generation with such a super-high resolution of $2048 \times 2048$.

Table C: Accelerating image generation on MS-COCO for the PIXART-$\Sigma$.

| Method | T | CLIP↑ | FID↓ | sFID↓ | CUR(%)↑ | Latency(s)↓ |
|---|---|---|---|---|---|---|
| PIXART-$\Sigma$ $512 \times 512$ (`cfg = 4.5`) | | | | | | |
| DPM-Solver++ (Lu et al., 2022b) | 20 | 31.20 | 26.81 | 42.79 | - | 1.912 |
| DPM-Solver++ (Lu et al., 2022b) | 15 | 31.23 | 25.99 | 42.08 | - | $1.435_{(\times 1.34)}$ |
| HarmoniCa | 20 | **31.31** | **24.30** | **42.73** | 65.43 | $\mathbf{1.206}_{(\times 1.59)}$ |
| PIXART-$\Sigma$ $1024 \times 1024$ (`cfg = 4.5`) | | | | | | |
| DPM-Solver++ (Lu et al., 2022b) | 20 | 31.37 | 20.98 | 27.47 | - | 9.467 |
| DPM-Solver++ (Lu et al., 2022b) | 15 | 31.34 | 21.63 | 28.68 | - | $7.100_{(\times 1.33)}$ |
| HarmoniCa | 20 | **31.36** | **20.94** | **27.25** | 59.52 | $\mathbf{6.432}_{(\times 1.47)}$ |
| PIXART-$\Sigma$ $2048 \times 2048$ (`cfg = 4.5`) | | | | | | |
| DPM-Solver++ (Lu et al., 2022b) | 20 | 31.19 | 23.61 | 51.12 | - | 14.198 |
| DPM-Solver++ (Lu et al., 2022b) | 15 | 31.26 | 24.40 | 53.34 | - | $9.782_{(\times 1.45)}$ |
| HarmoniCa | 20 | **31.36** | **23.88** | **53.25** | 58.29 | $\mathbf{9.410}_{(\times 1.51)}$ |

## F COMBINATION WITH QUANTIZATION

In this section, we conduct experiments to show the high compatibility of our HarmoniCa with the model quantization technique. In Tab. D, our method boosts a considerable speedup ratio from $\times 1.18$ to $\times 1.77$ with only a 0.16 FID increase for PIXART-$\alpha$ $256 \times 256$. In the future, we will explore combining our HarmoniCa with other acceleration techniques, such as pruning and distillation, to further reduce the computational demands for DiT.

## G EXPERIMENTAL DETAILS FOR QUANTIZATION

In Sec. F, we employ 8-bit channel-wise weight quantization and 8-bit layer-wise activation quantization for full-precision (FP32) DiT-XL/2 and half-precision (FP16) PIXART-$\alpha$. The former uses a 20-step DDIM sampler (Song et al., 2020a), while the latter employs a DPM-Solver++ sampler (Lu et al., 2022b) with the same steps. More specifically, we use MSE initialization (Nagel et al., 2021) for quantization parameters. For the quantization-aware fine-tuning stage, we set the learning rate of LoRA (Hu et al., 2021) and activation quantization parameters to $1e^{-6}$ and that of weight quantization parameters to $1e^{-5}$, respectively. Additionally, we employ 3.2K iterations for DiT-XL/2 (Peebles & Xie, 2023) and 9.6K iterations for PIXART-$\alpha$ (Chen et al., 2023) on a single NVIDIA H800

Table D: Results of the combination of our framework and an advanced quantization method: EfficientDM (He et al., 2024). IS↑ is for the former and CLIP↑ is for the latter in the table. Experimental details for quantization can be found in Sec. G. We mark the speedup ratio and the compression ratio in the brackets.

| Method | IS↑/CLIP↑ | FID↓ | sFID↓ | CUR(%)↑ | Latency(s)↓ | #Size(GB)↓ |
|---|---|---|---|---|---|---|
| DiT-XL/2 $256 \times 256$ (cfg = 1.5) | | | | | | |
| EfficientDM (He et al., 2024) | 172.70 | 6.10 | 4.55 | - | $0.591_{(\times 1.11)}$ | $0.64_{(\times 3.93)}$ |
| +HarmoniCa ($\beta = 4e^{-8}$) | 168.16 | 6.48 | 4.32 | 26.25 | $0.473_{(\times 1.40)}$ | $0.64_{(\times 3.93)}$ |
| PIXART-$\alpha$ $256 \times 256$ (cfg = 4.5) | | | | | | |
| EfficientDM (He et al., 2024) | 30.09 | 34.84 | 30.34 | - | $0.469_{(\times 1.18)}$ | $0.59_{(\times 1.98)}$ |
| +HarmoniCa | 30.23 | 35.00 | 31.38 | 53.34 | $0.301_{(\times 1.77)}$ | $0.59_{(\times 1.98)}$ |
| PIXART-$\alpha$ $512 \times 512$ (cfg = 4.5) | | | | | | |
| EfficientDM (He et al., 2024) | 30.71 | 25.82 | 41.64 | - | $0.461_{(\times 1.20)}$ | $0.59_{(\times 1.98)}$ |
| +HarmoniCa | 30.65 | 26.90 | 42.82 | 54.31 | $0.296_{(\times 1.80)}$ | $0.59_{(\times 1.98)}$ |

80G GPU. Other settings are the same as those from the original paper (He et al., 2024). Leveraging NVIDIA CUTLASS (Kerr et al., 2017) implementation, we evaluate the latency of quantized models employing the 8-bit multiplication for all the linear layers and convolutions.

# H    COMPARISON BETWEEN LEARNING-TO-CACHE AND HARMONICA WITH A LOW CUR(%)

In this section, we compare HarmoniCa with Learning-to-Cache (Ma et al., 2024b) at a relatively low CUR(%). As shown in Tab. E, both methods achieve a similar speedup ratio and even better performance than non-accelerated models. Therefore, we employ higher CUR in Tab. 2 to show our pronounced superiority.

Table E: Comparison results between Learning-to-Cache and HarmoniCa for the DiT-XL/2 with a low CUR(%).

| Method | T | IS↑ | FID↓ | sFID↓ | Prec.↑ | Recall↑ | CUR(%)↑ | Latency(s)↓ |
|---|---|---|---|---|---|---|---|---|
| DiT-XL/2 $256 \times 256$ (cfg = 1.5) | | | | | | | | |
| DDIM (Song et al., 2020a) | 20 | 224.37 | 3.52 | 4.96 | 78.47 | 58.33 | - | 0.658 |
| DDIM (Song et al., 2020a) | 15 | 214.77 | 4.17 | 5.54 | 77.43 | 56.30 | - | $0.564_{(\times 1.17)}$ |
| Learning-to-Cache (Ma et al., 2024b) | 20 | 228.19 | **3.49** | **4.66** | 79.32 | 59.10 | **22.05** | $\mathbf{0.545}_{(\times \mathbf{1.21})}$ |
| HarmoniCa ($\beta = 3e^{-8}$) | 20 | **228.79** | 3.51 | 4.76 | **79.43** | **59.32** | 21.07 | $0.547_{(\times 1.20)}$ |
| DiT-XL/2 $512 \times 512$ (cfg = 1.5) | | | | | | | | |
| DDIM (Song et al., 2020a) | 20 | 184.47 | 5.10 | 5.79 | 81.77 | 54.50 | - | 3.356 |
| DDIM (Song et al., 2020a) | 18 | 180.06 | 5.62 | 6.13 | 81.37 | 53.90 | - | $3.021_{(\times 1.11)}$ |
| Learning-to-Cache (Ma et al., 2024b) | 20 | 183.57 | 5.45 | 6.05 | **82.10** | 54.90 | 14.64 | $2.927_{(\times 1.15)}$ |
| HarmoniCa ($\beta = 2e^{-8}$) | 20 | **183.71** | **5.32** | **5.84** | 81.83 | **55.80** | 16.61 | $\mathbf{2.863}_{(\times \mathbf{1.17})}$ |

# I    COMPARISON BETWEEN $\Delta$-DIT AND HARMONICA

In this section, we compare HarmoniCa with $\Delta$-DiT (Chen et al., 2024b). Given that the code and implementation details of $\Delta$-DiT [11] are not open source, we report results derived from the original paper. Additionally, we evaluate performance sampling 5000 images as used in that study. As depicted in Tab F, our framework further decreases 20% latency and gains 3.52 IS improvement compared with $\Delta$-DiT for PIXART-$\alpha$ with a 20-step DPM-Solver++ sampler (Lu et al., 2022b).

---

[11] $\Delta$-DiT presents the speedup ratio based on multiply-accumulate operates (MACs). Here we report the results according to the latency in that study.

Table F: Comparison results between $\Delta$-DiT and HarmoniCa on on MS-COCO for PIXART-$\alpha$ 1024 $\times$ 1024.

| Method | T | CLIP↑ | FID↓ | IS↑ | CUR(%)↑ | Speedup↑ |
|---|---|---|---|---|---|---|
| PIXART-$\alpha$ 1024 $\times$ 1024 (`cfg` = 4.5) | | | | | | |
| DPM-Solver++ (Lu et al., 2022b) | 20 | 31.07 | 31.98 | 41.30 | - | - |
| DPM-Solver++ (Lu et al., 2022b) | 13 | 31.04 | 33.29 | 39.15 | - | ×1.54 |
| $\Delta$-DiT (Chen et al., 2024b) | 20 | 30.40 | 35.88 | 32.22 | 37.49 | ×1.49 |
| HarmoniCa ($\beta = 1e^{-3}$) | 20 | **31.08** | **32.97** | **40.67** | **62.31** | **×1.63** |

## J COMPARISON BETWEEN LEARNING-TO-CACHE WITH DIFFERENT SAMPLING STRATEGIES

For the implementation details [12], Learning-to-Cache uniformly samples an even timestep $t$ during each training iteration [13], as opposed to sampling any timestep from the set $\{1, \ldots, T\}$ as mentioned in Alg. 1 of its original paper. Consequently, according to Fig. 3, only $r_{t,i}$, where $t$ is an odd timestep, is learnable, while the remaining values are set to one. We compare Learning-to-Cache under different sampling strategies (*i.e.*, sampling an even timestep or without this constraint for each training iteration) against HarmoniCa. As shown in Tab. G, our framework—whether training the entire `Router` or only parts of it (similar to the Learning-to-Cache implementation)—consistently outperforms Learning-to-Cache regardless of the sampling strategy.

It should be noted that the experiments in Sec. 5, with the exception of those in Tab. 4, use an implementation that uniformly samples an even timestep $t$ during each training iteration. This approach achieves significantly higher performance compared to sampling without constraints.

Table G: Comparison results between Learning-to-Cache with different sampling strategies and HarmoniCa for the DiT-XL/2 256 $\times$ 256. "♣" denotes that only parts of the `Router` corresponding to odd timesteps are learnable and the remaining values are set to one (*i.e.*, disable reusing cached features).

| Method | T | IS↑ | FID↓ | sFID↓ | Prec.↑ | Recall↑ | CUR(%)↑ | Latency(s)↓ |
|---|---|---|---|---|---|---|---|---|
| DiT-XL/2 256 $\times$ 256 (`cfg` = 1.5) | | | | | | | | |
| DDIM (Song et al., 2020a) | 20 | 224.37 | 3.52 | 4.96 | 78.47 | 58.33 | - | 0.658 |
| Learning-to-Cache (Ma et al., 2024b) | 20 | 115.00 | 18.57 | 16.18 | 60.35 | 62.98 | 32.68 | 0.483$_{(×1.36)}$ |
| Learning-to-Cache♣ (Ma et al., 2024b) | 20 | 201.37 | 5.34 | 6.36 | 75.04 | 56.09 | 35.60 | 0.468$_{(×1.41)}$ |
| HarmoniCa♣ ($\beta = 3.5e^{-8}$) | 20 | 205.39 | **4.86** | 5.92 | 75.06 | 57.97 | 36.07 | 0.463$_{(×1.42)}$ |
| HarmoniCa | 20 | **206.57** | 4.88 | **5.91** | **75.20** | **58.74** | **37.50** | **0.456**$_{(×1.44)}$ |

## K COMPARISON BETWEEN HARMONICA AND ADDITIONAL CACHING-BASED METHODS

To highlight HarmoniCa's advantages, we compare it with DeepCache (Ma et al., 2024c) and Faster Diffusion (Li et al., 2023a) on a single A6000 GPU. Due to the partial open-sourcing of the compared methods and the lack of implementation details, we directly report their results from Learning-to-Cache. As shown in Tab. H, HarmoniCa achieves a minimal FID increase of less than 0.05, while providing a 1.65× speedup, outperforming both Faster Diffusion and DeepCache. Notably, DeepCache is constrained by the U-shaped structure, making it unsuitable for DiTs.

---

[12] Let $T$ be an even number here.

[13] `https://github.com/horseee/learning-to-cache/blob/main/DiT/train_router.py#L244-L247`

Table H: Comparison between different caching-based approaches. We use U-ViT (Bao et al., 2023) on ImageNet 256×256 here.

| Method | T | FID↓ | Latency(s)↓ |
|---|---|---|---|
| DPM-Solver (Lu et al., 2022a) | 20 | 2.57 | 7.60 |
| Faster Diffusion (Li et al., 2023a) | 20 | 2.82 | $5.95_{(\times 1.28)}$ |
| DeepCache (Ma et al., 2024c) | 20 | 2.70 | $4.68_{(\times 1.62)}$ |
| HarmoniCa | 20 | **2.61** | $\mathbf{4.60}_{(\times \mathbf{1.65})}$ |

## L  COMPARISON BETWEEN HARMONICA AND ADDITIONAL ACCELERATION METHODS

As shown in Tab. I, we compare our HarmoniCa with advanced quantization and pruning methods. Our method significantly outperforms these methods, demonstrating the substantial benefit of feature cache for accelerating DiT models. It is important to note that the speedup ratio for quantization is partially determined by hardware support which we do not rely on and the current customized CUDA kernel often lacks optimization on H800's Hopper architecture. Additionally, our method is orthogonal to these approaches, meaning it can be combined with them for further acceleration (results of EfficientDM + HarmoniCa have been presented in Sec. F). *We believe the significant performance drop of PTQ4DiT here results from a small-sampling-step DDIM sampler. A 50/250-step DDPM sampler is used in the original paper.*

**Experimental details:** We employ the bit-width of w8a8 for quantization. Specifically, the implementation details for EfficientDM can be found in Sec. G. For PTQ4DiT, we implemented the DDIM sampler and re-run the open-source code, which originally only supported DDPM. For Diff-pruning, we re-implement the method for the DiT model (which originally only supported U-Net models) and follow the settings specified in the original paper. For quantization, latency tests were conducted with the w8a8 multiplication from He et al. (2024).

Table I: Comparison between different acceleration approaches. We use DiT-XL/2 on ImageNet 256×256 here. "*" denotes the latency was tested on one A100 GPU.

| Method | T | IS↑ | FID↓ | sFID↓ | Latency(s)↓ | Latency(s)↓* |
|---|---|---|---|---|---|---|
| DDIM (Zhang et al., 2022) | 20 | 224.37 | 3.52 | 4.96 | 0.658 | 1.217 |
| EfficientDM (He et al., 2024) | 20 | 172.70 | 6.10 | 4.55 | $0.591_{(\times 1.11)}$ | $0.842_{(\times 1.45)}$ |
| PTQ4DIT (Wu et al., 2024) | 20 | 17.06 | 71.82 | 23.16 | $0.577_{(\times 1.14)}$ | $0.839_{(\times 1.45)}$ |
| Diff-pruning (Fang et al., 2023) | 20 | 168.10 | 8.22 | 6.20 | $0.458_{(\times 1.44)}$ | $\mathbf{0.813}_{(\times \mathbf{1.50})}$ |
| HarmoniCa | 20 | **206.57** | **4.88** | **5.91** | $\mathbf{0.456}_{(\times \mathbf{1.44})}$ | $0.815_{(\times 1.49)}$ |

## M  ADDITIONAL METRICS FOR THE IMAGE-ERROR PROXY $\lambda^{(t)}$

As shown in Tab. J, under the same speedup ratio, we further test MS-SSIM (Wang et al., 2003) and LPIPS (Zhang et al., 2018) (AlexNet (Krizhevsky et al., 2017) to extract image features) which are designed to evaluate natural image quality as metrics for $\lambda^{(t)}$. These metrics exhibit comparable performance compared with $\|\cdot\|_F^2$. For instance, LPIPS slightly outperforms in FID and sFID, while $\|\cdot\|_F^2$ marginally excels in IS.

Table J: Effect of additional different metrics for $\lambda^{(t)}$. We use DiT-XL/2 on ImageNet 256×256 with a 20-step DDIM sampler here.

| $\lambda^{(t)}$ | $\|x_0 - x_0^{(t)}\|_F^2$ | $1 - \text{MS-SSIM}(x_0, x_0^{(t)})$ | $\text{LPIPS}(x_0, x_0^{(t)})$ |
|---|---|---|---|
| IS↑ | **206.57** | 204.72 | 205.83 |
| FID↓ | 4.88 | 4.91 | **4.83** |
| sFID↓ | 5.91 | 5.83 | **5.57** |
| CUR(%)↑ | **37.50** | 37.68 | 37.32 |
| Latency↓ | $\mathbf{0.456}_{(\times \mathbf{1.44})}$ | $\mathbf{0.456}_{(\times \mathbf{1.44})}$ | $\mathbf{0.456}_{(\times \mathbf{1.44})}$ |

## N   APPLY THE TRAINED ROUTER TO A DIFFERENT SAMPLER FROM TRAINING DURING INFERENCE

As shown in Tab. K, the `Router` trained with one diffusion sampler can indeed be applied to a different sampler, such as DPM-Solver++→Sa-Solver (6th row) and IDDPM→DPM-Solver++ (10th row). However, the performance of these trials is much worse than the standard HarmoniCa. We believe this is due to the discrepancies in sampling trajectories and noise scheduling between the two samplers, which need to be accounted for during the `Router` training. In other words, the sampler used for training should match the one used during inference to improve the performance.

Table K: Results of applying the trained `Router` to a different sampler from training during inference. "A→B" denotes the `Router` trained with the sampler "A" is directly used during inference with the sampler "B".

| Method | T | CLIP↑ | FID↓ | sFID↓ | CUR(%)↑ | Latency(s)↓ |
|---|---|---|---|---|---|---|
| PIXART-$\alpha$ 256 × 256 (`cfg` = 4.5) | | | | | | |
| SA-Solver (Xue et al., 2024) | 20 | 31.28 | 23.96 | 35.63 | - | 0.677 |
| SA-Solver (Xue et al., 2024) | 16 | 31.16 | 26.27 | 39.28 | - | $0.520_{(\times 1.30)}$ |
| HarmoniCa | 20 | **31.23** | **24.17** | **35.98** | 42.12 | $\mathbf{0.516}_{(\times \mathbf{1.31})}$ |
| HarmoniCa (DPM-Solver++→ SA-Solver) | 20 | 31.18 | 25.99 | 37.94 | 40.98 | $0.523_{(\times 1.29)}$ |
| DPM-Solver++ (Lu et al., 2022b) | 100 | 31.30 | 25.01 | 35.42 | - | 2.701 |
| DPM-Solver++ (Lu et al., 2022b) | 73 | 31.27 | 25.16 | 36.11 | - | $2.005_{(\times 1.35)}$ |
| HarmoniCa | 100 | **31.35** | **24.96** | **35.19** | 51.89 | $\mathbf{1.998}_{(\times \mathbf{1.35})}$ |
| HarmoniCa (IDDPM→DPM-Solver++) | 100 | 31.22 | 25.43 | 39.84 | 50.98 | $2.002_{(\times 1.35)}$ |

## O   PERFORMANCE COMPARISON WITH THE INCREASE OF THE SPEEDUP RATIO

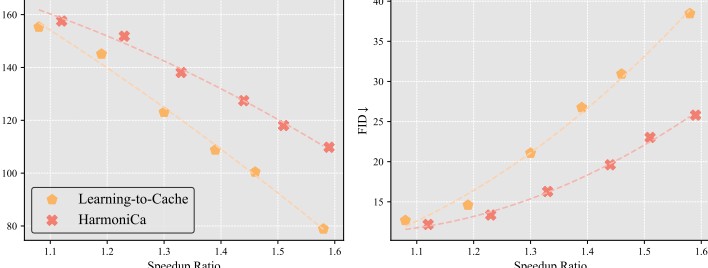

Figure B: IS/FID with the increase of the speedup ratio for different methods. We employ DiT-XL/2 with a 10-step DDIM sampler on ImageNet 256 × 256.

To emphasize the significant advantage of our method over Learning-to-Cache, we present the IS and FID results as the speedup ratio increases for both Learning-to-Cache and our HarmoniCa in Fig. B. As the speedup ratio grows, the gap between Learning-to-Cache and our approach widens substantially. Specifically, with a speedup ratio of approximately 1.6, HarmoniCa achieves substantially higher IS and lower FID scores, 30.90 and 12.34, respectively, compared to Learning-to-Cache. Furthermore, our method consistently outperforms Learning-to-Cache across all speedup ratios.

## P   ADDITIONALLY RESULTS OF HARMONICA WITH SA-SOLVER

Regarding the comparison with SA-Solver, we conducted additional experiments to highlight HarmoniCa's advantages. In Tab. K, we use fewer denoising steps (20 steps, compared to 25 in the main texts). With a similar latency, our method outperforms the 16-step Sa-Solver by 2.10 FID and 3.30 sFID (4th row *vs.* 5th row). In Tab. L, we test our method with higher resolutions. As resolution increases, HarmoniCa delivers more pronounced benefits than the fewer-step Sa-Solver. Specifically, HarmoniCa achieves lower FID and sFID, and a higher CLIP score with a $1.46\times$ speedup over the non-accelerated model. In contrast, the 20-step Sa-Solver performs worse than the non-accelerated model, with a $1.30\times$ speedup.

Table L: HarmoniCa +SA-Solver for high resolution image generation on MS-COCO captions.

| Method | T | CLIP↑ | FID↓ | sFID↓ | CUR(%)↑ | Latency(s)↓ |
|---|---|---|---|---|---|---|
| PixArt-$\alpha$ 512 × 512 (`cfg = 4.5`) | | | | | | |
| SA-Solver (Xue et al., 2024) | 25 | 31.23 | 25.43 | 39.84 | - | 2.263 |
| SA-Solver (Xue et al., 2024) | 20 | 31.19 | 25.85 | 40.08 | - | 1.738$_{(\times 1.30)}$ |
| HarmoniCa | 25 | **31.24** | **24.44** | 39.87 | 52.04 | **1.611**$_{(\times 1.40)}$ |
| PixArt-$\alpha$ 1024 × 1024 (`cfg = 4.5`) | | | | | | |
| SA-Solver (Xue et al., 2024) | 25 | 31.05 | 23.65 | 38.12 | - | 11.931 |
| SA-Solver (Xue et al., 2024) | 20 | 31.02 | 23.88 | 39.41 | - | 9.209$_{(\times 1.30)}$ |
| Harmonica | 25 | **31.10** | **23.52** | **37.89** | 52.46 | **8.151**$_{(\times 1.46)}$ |

# Q    RESULTS OF T2I GENERATION ON ADDITIONAL DATASETS AND METRICS

Table M: Accelerating image generation on MJHQ-30K (Li et al., 2024a) and sDCI (Urbanek et al., 2024) for the PIXART-$\alpha$. We sample 30K images for MJHQ-30K and 5K images for sDCI. "IR" denotes Image Reward.

| Method | T | MJHQ | | | | | sDCI | | | | | Latency (s)↓ |
|---|---|---|---|---|---|---|---|---|---|---|---|---|
| | | Quality | | | Similarity | | Quality | | | Similarity | | |
| | | FID↓ | IR↑ | CLIP↑ | LPIPS↓ | PSNR↑ | FID↓ | IR↑ | CLIP↑ | LPIPS↓ | PSNR↑ | |
| PixArt-$\alpha$ 512 × 512 (`cfg = 4.5`) | | | | | | | | | | | | |
| DPM-Solver | 20 | 7.04 | 0.947 | 26.04 | - | - | 11.47 | 0.994 | 25.22 | - | - | 1.759 |
| DPM-Solver | 15 | 7.45 | 0.899 | 26.02 | 0.138 | 21.41 | 11.55 | 0.876 | 25.19 | 0.178 | 19.85 | 1.291$_{(\times 1.36)}$ |
| HarmoniCa | 20 | **7.01** | **0.955** | **26.04** | **0.129** | **22.09** | **11.49** | **0.951** | **25.22** | **0.171** | **20.01** | **1.168**$_{(\times 1.51)}$ |
| PixArt-$\alpha$ 1024 × 1024 (`cfg = 4.5`) | | | | | | | | | | | | |
| DPM-Solver | 20 | 6.24 | 0.966 | 26.23 | - | - | 10.96 | 0.986 | 25.56 | - | - | 9.470 |
| DPM-Solver | 15 | 6.49 | 0.921 | 26.18 | 0.107 | 23.98 | 11.22 | 0.942 | 25.51 | 0.186 | 18.44 | 7.141$_{(\times 1.32)}$ |
| HarmoniCa | 20 | **6.31** | **0.944** | **26.21** | **0.101** | **25.01** | **11.09** | **0.979** | **25.54** | **0.175** | **20.42** | **6.289**$_{(\times 1.51)}$ |

In addition to the evaluations on ImageNet and MS-COCO, we conducted further tests using the high-quality MJHQ-30K (Li et al., 2024a) and sDCI (Urbanek et al., 2024) datasets with PixArt-$\alpha$ models. We added several metrics, including Image Reward (Xu et al., 2024), LPIPS (Learned Perceptual Image Patch Similarity) (Zhang et al., 2018), and PSNR (Peak Signal-to-Noise Ratio). The results, summarized in the following table, demonstrate that HarmoniCa consistently outperforms DPM-Solver across all metrics on both the MJHQ and sDCI datasets. For instance, at the 512×512 resolution, HarmoniCa achieves an FID of 7.01 on the MJHQ dataset, which is lower than the 7.04 FID of DPM-Solver with 20 steps, indicating better image quality. Additionally, under the same configuration, HarmoniCa achieves a PSNR of 22.09, compared to DPM-Solver's 21.41 with 15 steps, reflecting better numerical similarity.

## R SENSITIVITY OF HARMONICA TO THE VALUE OF THE THRESHOLD $\tau$

We conduct an ablation study on different values of the caching threshold $\tau \in [0, 1)$, as shown in Tab. N. The results demonstrate that HarmoniCa is robust *w.r.t* variations in $\tau$.

Table N: Performance of HarmoniCa across different values of $\tau \in [0, 1)$ ($\tau$ is the `router` threshold as described in Sec. 3). We employ DiT-XL/2 on ImageNet $256 \times 256$ here.

| $\tau$ | T | IS↑ | FID↓ | sFID↓ | Latency(s)↓ |
|---|---|---|---|---|---|
| 0.1 | 10 | 151.83 | 13.35 | 11.13 | $0.270_{(\times 1.23)}$ |
| 0.5 | 10 | 151.80 | 13.41 | 11.09 | $0.269_{(\times 1.23)}$ |
| 0.9 | 10 | 151.78 | 13.37 | 11.08 | $0.270_{(\times 1.23)}$ |

## S QUALITATIVE COMPARISON & ANALYSES

As shown in Fig. C and D, we provide qualitative comparison between HarmoniCa and other baselines, *e.g.*, Learning-to-Cache (Ma et al., 2024b), FORA (Selvaraju et al., 2024), and the fewer-step sampler. Our HarmoniCa with a higher speedup ratio can generate more accurate details, *e.g., 2nd column of Fig. D (d) vs. (b)* and objective-level traits, *e.g., 2nd column of Fig. C (d) vs. (c).*

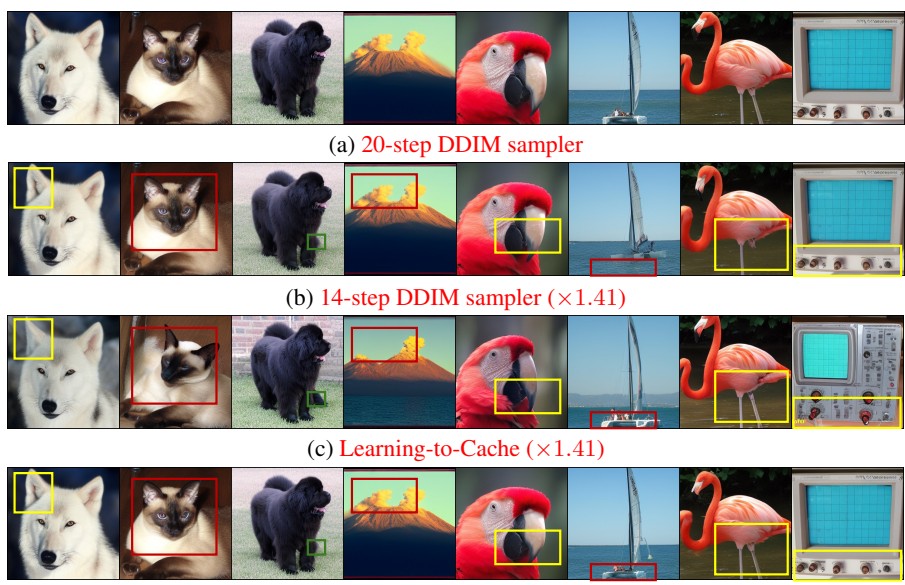

(a) 20-step DDIM sampler

(b) 14-step DDIM sampler (×1.41)

(c) Learning-to-Cache (×1.41)

(d) HarmoniCa (×1.44)

Figure C: Random samples from DiT-XL/2 $256 \times 256$ (Chen et al., 2023) with different acceleration methods. The resolution of each sample is $256 \times 256$. We employ `cfg = 4` here for better visual results. Key differences are highlighted using rectangles with various colors.

## T VISUALIZATION RESULTS

As demonstrated in Figures E to K, we present random samples from both the non-accelerated DiT models and ones equipped with HarmoniCa, using a fixed random seed. Other settings are the same as mentioned in the former experiments. Our approach not only significantly accelerates inference but also produces results that closely resemble those of the original model. For a detailed comparison, zoom in to closely examine the relevant images.

*"Hello kitty cake surrounded by strawberries and kukohon, in the style of camille pissarro, yanjun cheng, bella kotak, iso 200, oshare kei, caninecore, meticulous design"*

*"The small fluffy Corgi is sitting among some flowers and mountains background , in the style of matte painting, happy facial expression, gongbi, white and cyan, movie still, eye catching detail, textured shading culture infused"*

*"Landscape photography, clean sharp focus, hyperrealist photography, real photography, wide full body angle, editorial, luxury reort pool in Positano, Italy, 24mm Kodak film. dramatic backlighting, sunny, bright, vibrant and colorful, soft body, portra 800 ISO, medium format grain, realistic, sharp focus, vintage feel"*

*"Simple 8bit pixel art, an astronomical observatory with open dome slit on the peak of a mountain at dusk, lit by the glow of stars and planets emerging at the retreat of the setting sun, a beautiful landscape on the valley below, in the style of video game detailed 8bit pixel art, photography"*

(a) 20-step DPM-Solver

(b) 15-step DPM-Solver (×1.36)

(c) FORA (×1.34)

(d) HarmoniCa (×1.51)

Figure D: Random samples from PIXART-α 512×512 (Chen et al., 2023) with different acceleration methods. The resolution of each sample is 512 × 512.

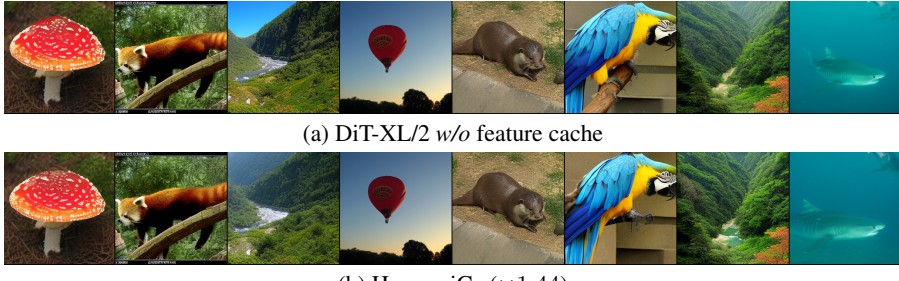

(a) DiT-XL/2 *w/o* feature cache

(b) HarmoniCa (×1.44)

Figure E: Random samples from (a) non-accelerated and (b) accelerated DiT-XL/2 256 × 256 (Chen et al., 2023) with a 20-step DDIM sampler (Song et al., 2020a). The resolution of each sample is 256 × 256. We mark the speedup ratio in the brackets.

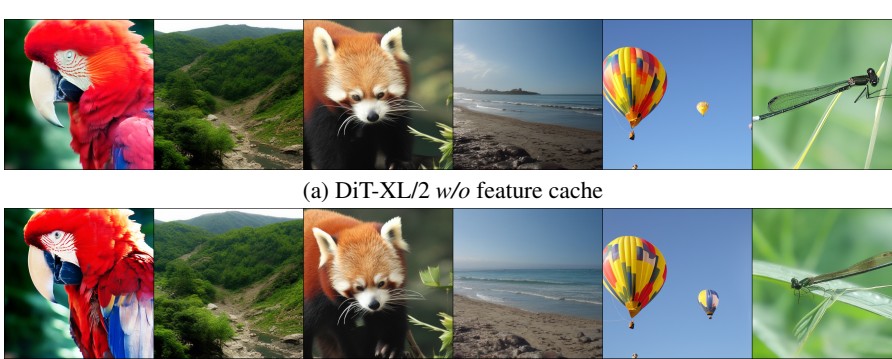

(a) DiT-XL/2 *w/o* feature cache

(b) HarmoniCa (×1.30)

Figure F: Random samples from (a) non-accelerated and (b) accelerated DiT-XL/2 $512 \times 512$ (Chen et al., 2023) with a 20-step DDIM sampler (Song et al., 2020a). The resolution of each sample is $512 \times 512$.

> "A cozy wooden cabin perched on the side of a mountain, overlooking a vast valley. The sun is setting, casting a golden glow over the cabin and the surrounding landscape. Smoke rises from the chimney, and the scene feels warm and inviting."

> "A dense forest at night, illuminated by the glow of the full moon. Fireflies dance in the air, creating soft, flickering lights among the trees. The forest floor is covered in moss and ferns, and the scene feels magical and tranquil."

> "A peaceful oasis in the middle of an endless desert, with palm trees reflecting in the crystal-clear water. The early morning sun is rising, casting a golden glow over the sand dunes, while the sky transitions from deep blue to vibrant orange."

> "An old, grand Victorian library with tall bookshelves filled with leather-bound books. Sunlight streams in through large stained-glass windows, casting colorful patterns on the floor. A sense of history and knowledge fills the air."

> "A nighttime scene of a festival where hundreds of glowing lanterns float down a river, their warm light reflecting on the water. People stand on the banks, watching the lanterns drift by, with fireworks lighting up the sky above."

(a) PIXART-$\alpha$ *w/o* feature cache

(b) HarmoniCa (×1.52)

Figure G: Random samples from (a) non-accelerated and (b) accelerated PIXART-$\alpha$ $256 \times 256$ (Chen et al., 2023) with a 20-step DPM-Solver++ sampler (Lu et al., 2022b). The resolution of each sample is $256 \times 256$. Text prompts are exhibited above the corresponding images

*"A floating crystal palace high above the clouds, with intricate spires and towers made of transparent, glowing crystals. The sky is filled with radiant sunlight, and the clouds below reflect the palace's brilliance, creating a heavenly, magical scene."*

*"Two samurais clad in futuristic, neon-infused armor face off in a high-tech dojo. Their glowing katanas clash as electric sparks fly. The scene is set against a backdrop of towering city buildings and a bright, cyberpunk night sky."*

*"A massive dragon with shimmering scales glides over a dense, enchanted forest. Its wings create powerful gusts of wind, rustling the treetops below. The dragon's scales reflect the sunlight, creating a dazzling, majestic spectacle."*

*"In the depths of a dark, shadowy forest, a glowing portal of swirling blue and purple energy opens between ancient, twisted trees. A faint light emanates from the portal, casting an otherworldly glow on the forest floor covered in fallen leaves and mist."*

(a) PIXART-$\alpha$ *w/o* feature cache

(b) HarmoniCa ($\times 1.51$)

Figure H: Random samples from (a) non-accelerated and (b) accelerated PIXART-$\alpha$ $512\times512$ (Chen et al., 2023) with a 20-step DPM-Solver++ sampler (Lu et al., 2022b). The resolution of each sample is $512 \times 512$.

*"A medieval knight in full armor standing in a castle courtyard, holding a sword with both hands. His face is solemn as he prepares for battle, while the flags of the kingdom flutter behind him in the wind."*

*"A ballet dancer mid-pirouette on an empty stage, her elegant movements illuminated by a single spotlight. Her tutu swirls around her as she leaps gracefully through the air, capturing the essence of motion and grace."*

*"An ancient, majestic castle nestled atop a mountain peak, surrounded by swirling clouds, illuminated by golden sunlight. A dragon circles above, while knights stand guard below. The scene is full of magical realism, detailed stone walls, and elaborate banners flapping in the wind."*

*"A futuristic space station orbiting a colorful planet, surrounded by glowing stars and nebulae. Astronauts float near the station, with sleek spacecraft docking. The image captures the vastness and wonder of space, with intricate details on the station's metallic structure."*

*"A curious red fox exploring a snow-covered forest, its fur blending with the white landscape. Its sharp eyes scan the surroundings as it sniffs the ground, leaving delicate paw prints in the snow."*

*"A sleek, advanced city at dawn, with shimmering glass towers, floating gardens, and high-tech transportation systems. The sky is painted with pastel colors as the sun rises, casting a golden glow over the futuristic landscape."*

(a) PIXART-$\alpha$ *w/o* feature cache

(b) HarmoniCa ($\times 1.51$)

Figure I: Random samples from (a) non-accelerated and (b) accelerated PIXART-$\alpha$ $1024 \times 1024$ (Chen et al., 2023) with a 20-step DPM-Solver++ sampler (Lu et al., 2022b). The resolution of each sample is $1024 \times 1024$.

*"Two colossal mechas, each covered in intricate armor plating and glowing power cores, engage in battle in the middle of a futuristic city. Skyscrapers crumble around them as they exchange powerful blows, and the energy radiating from their weapons lights up the night sky."*

*"A gargantuan sea creature with towering spines and glowing eyes rises from the ocean, water cascading off its massive form. Lightning illuminates the stormy sky as ships scramble to escape its wrath, emphasizing the creature's immense size and power."*

*"A colossal, ancient citadel made of shining marble and gold, perched atop the clouds. Massive towers and archways reach towards a sky filled with radiant sunlight, while a staircase of light descends from the heavens, hinting at the citadel's divine origins."*

*"A vast army of warriors clad in glistening armor charging across an icy battlefield under a stormy, dark sky. Blizzards rage around them, and the ground shakes as they clash with their enemies. The scene is filled with motion, energy, and the raw power of war."*

*"A majestic phoenix, its wings spread wide, emerges from a massive pillar of fire. The flames swirl around it in a dance of red, gold, and blue, while sparks and embers fill the air. Its form is both terrifying and beautiful, a symbol of rebirth and eternal power."*

*"A titanic clash between two massive, glowing deities in the sky, with thunderbolts and energy waves exploding around them. Below, mountains crumble and oceans churn as their power shakes the very fabric of reality, creating a breathtaking cosmic spectacle."*

(a) PIXART-Σ *w/o* feature cache

(b) HarmoniCa (×1.47)

Figure J: Random samples from (a) non-accelerated and (b) accelerated PIXART-Σ 1024 × 1024 (Chen et al., 2024a) with a 20-step DPM-Solver++ sampler (Lu et al., 2022b). The resolution of each sample is 1024 × 1024.

*"Two samurais locked in a fierce duel under a cherry blossom tree, depicted in the traditional Japanese Ukiyo-e style. The bold outlines, flat colors, and exaggerated poses capture the intensity of the moment, while the delicate cherry blossoms fall gently around them."*

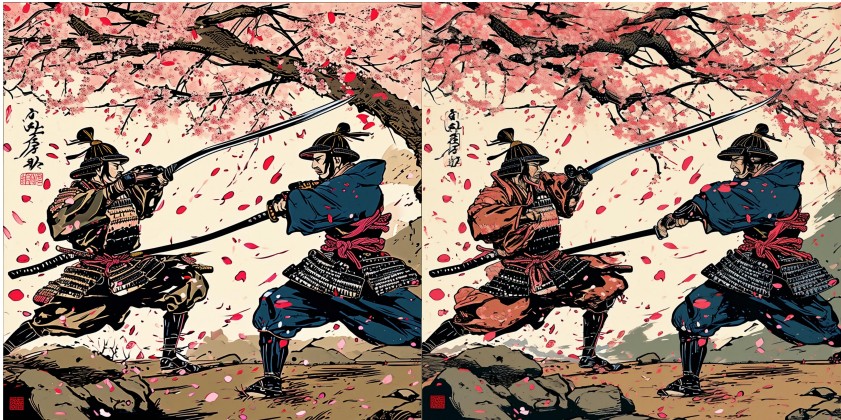

*"A peaceful alpine village nestled in the shadow of towering, snow-capped mountains, painted in a detailed realism oil painting style. The wooden houses have sloping roofs covered in snow, and smoke rises gently from their chimneys. The brushwork captures the texture of the wood and the soft shadows cast by the evening light."*

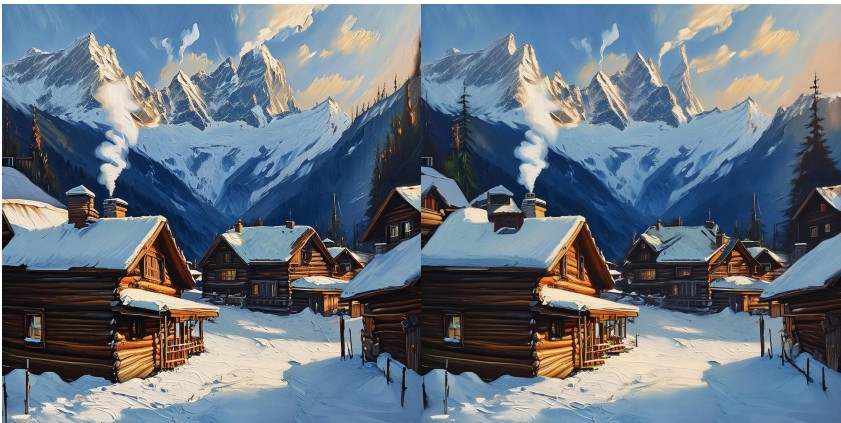

*"A studio photograph of an elegant Asian woman in a flowing silk dress. Her hair is styled in soft waves, and the smooth fabric of her dress reflects the studio lights gently. The high-definition shot focuses on the intricate textures of her skin and hair, as well as the subtle glint of light in her eyes."*

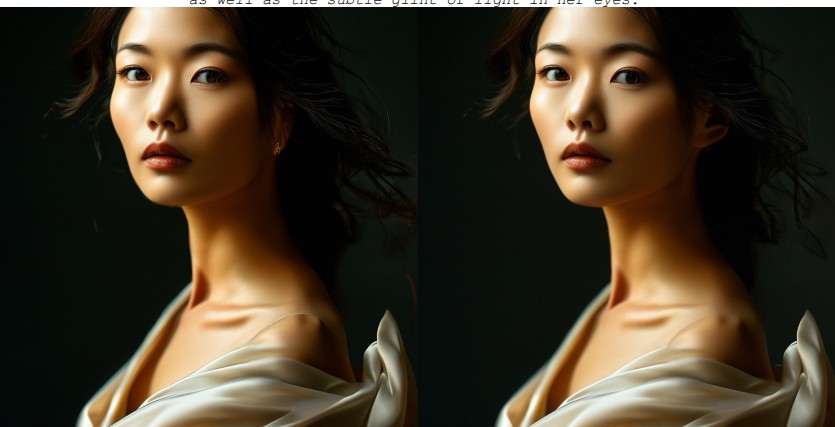

(a) PIXART-$\Sigma$ *w/o* feature cache        (b) HarmoniCa ($\times 1.51$)

Figure K: Random samples from (Left) non-accelerated and (Right) accelerated PIXART-$\Sigma$-2K (Chen et al., 2024a) with a 20-step DPM-Solver++ sampler (Lu et al., 2022b). The resolution of each sample is $2048 \times 2048$.

