# OpenReview forum: "HarmoniCa: Harmonizing Training and Inference for Better Feature Cache in Diffusion Transformer Acceleration"
_ICLR.cc/2025/Conference — Submitted to ICLR 2025_

### Official Review · Reviewer_UrSw · 2024-10-27

**Soundness:** 2
**Presentation:** 2
**Contribution:** 2
**Rating:** 8
**Confidence:** 1

**Summary:**

The paper titled "HarmoniCa: Harmonizing Training and Inference for Better Feature Cache in Diffusion Transformer Acceleration" proposes a method to enhance the efficiency of Diffusion Transformers (DiTs) in generative models by harmonizing the training and inference phases. DiTs, while powerful in generative tasks, face high inference costs due to redundant computations. HarmoniCa addresses this through a novel feature caching framework that includes two key components:

1. **Step-Wise Denoising Training (SDT)**: This component aligns the training process with the inference phase by maintaining continuity in the denoising trajectory, allowing the model to effectively reuse cached features across timesteps, as it would in inference.

2. **Image Error Proxy-Guided Objective (IEPO)**: IEPO incorporates a proxy mechanism that approximates the final image error caused by cached feature reuse. This helps balance image quality with efficient cache usage, bridging the gap between training objectives and inference goals.

The paper demonstrates that HarmoniCa significantly reduces training time (by around 25%) and achieves faster inference with improved image quality compared to non-cached and traditional learning-based methods. Through extensive experiments, HarmoniCa shows promising speedup ratios and better performance across multiple models and resolutions, highlighting its effectiveness and applicability in high-resolution generative tasks.

**Strengths:**

### Strengths of the Paper

The paper, "HarmoniCa: Harmonizing Training and Inference for Better Feature Cache in Diffusion Transformer Acceleration," offers a robust contribution to optimizing Diffusion Transformers (DiTs) for generative tasks through innovative feature caching techniques. Below, I evaluate the paper’s strengths across originality, quality, clarity, and significance.

---


The HarmoniCa framework addresses longstanding challenges in generative model efficiency by rethinking the training-inference dichotomy in Diffusion Transformers. This originality stems from its introduction of **Step-Wise Denoising Training (SDT)** and the **Image Error Proxy-Guided Objective (IEPO)**, both of which directly tackle the inefficiencies introduced by common feature caching methods. HarmoniCa’s approach to caching involves harmonizing the training and inference phases rather than optimizing each separately. This unique alignment of caching goals across both phases is not only original but also has substantial implications for generative model performance.
HarmoniCa’s emphasis on bridging the gap between training and inference caching, specifically addressing **Prior Timestep Disregard** and **Objective Mismatch**, reframes typical caching strategies by focusing on end-to-end generative quality. The **IEPO** component, which approximates final image error using a proxy, is a novel way to reconcile training objectives with final image quality, enabling more practical feature reuse in diffusion models.

---
The paper’s methodological rigor is evident in its clear experimental protocols and validation against state-of-the-art methods. The authors provide comprehensive experiments across multiple high-resolution models and resolutions, including **class-conditional** and **text-to-image (T2I)** tasks. HarmoniCa’s performance gains (e.g., 1.5× speedup with no significant loss in image quality) underscore the quality of the proposed solution. Additionally, the experiments carefully balance different cache utilization ratios (CUR), demonstrating HarmoniCa’s adaptability to varying model requirements. Extensive comparisons with current methods like **Learning-to-Cache** and **FORA** show that HarmoniCa consistently improves upon existing feature cache methods, providing concrete evidence for its efficacy and robustness. The inclusion of detailed **ablation studies** on the impact of IEPO and SDT further reinforces the work’s quality. These studies examine the effects of cache ratios, loss coefficients, and proxy metrics, showcasing a thorough understanding of the parameters that influence HarmoniCa’s success.

---
The paper is well-organized, with each section systematically addressing a specific aspect of the proposed solution: **Introduction and Background** sections clearly frame the limitations of existing methods, paving the way for the presentation of HarmoniCa as a targeted solution. **Methodology** is presented with sufficient depth, balancing detailed technical explanations and visual aids. For instance, **Figure 3** and **Figure 4** effectively illustrate how SDT and IEPO integrate into the DiT framework, making the mechanisms of HarmoniCa accessible even to readers less familiar with diffusion model architectures. **Algorithmic Details**: The paper includes equations and structured pseudo-code, enhancing reproducibility and understanding of HarmoniCa’s inner workings.  the authors provide context for critical terms (e.g., CUR, SDT), allowing readers to grasp the innovative elements without requiring extensive prior knowledge in feature caching.

---
The HarmoniCa framework represents a substantial step forward in making high-resolution generative models more practical by **lowering inference and training costs**. Its significance extends to several dimensions: **Practical Utility**: By reducing training time by ~25% and inference latency by ~1.5×, HarmoniCa enables more efficient deployment of generative models in real-world applications, such as **text-to-image synthesis** and **class-conditional generation**. These improvements address critical barriers to the broader adoption of diffusion models in industry.  **Broad Applicability**: The framework’s adaptability across multiple architectures and resolution settings underscores its broad utility. The ability to apply HarmoniCa without image data during training expands its applicability to scenarios with limited training resources.  **Scientific Advancement**: This paper advances research in efficient generative modeling by proposing an original, learning-based caching strategy that can inform future developments in DiTs and potentially inspire similar approaches in other transformer-based generative models.

**Weaknesses:**

- The HarmoniCa framework’s reliance on multiple components (SDT, IEPO, and the custom cache router) could limit its accessibility and reproducibility for practitioners, especially those with limited computational resources. While the model is described in detail, implementing and integrating SDT and IEPO into an existing Diffusion Transformer may pose a significant challenge due to the additional computational and architectural overhead.

 - While HarmoniCa is tested across multiple models and resolutions, it is evaluated only on **ImageNet** (for class-conditioned tasks) and **MS-COCO** (for text-to-image tasks). These datasets are popular benchmarks but may not represent the variety of distributions found in real-world data, especially in domains like video generation or multi-modal applications where diffusion models are increasingly applied.

 - The **Image Error Proxy-Guided Objective (IEPO)** is a central contribution but relies on approximating final image quality based on intermediate denoising steps. The validity of this proxy could vary significantly depending on the task and model architecture, yet the paper provides limited empirical analysis of IEPO’s sensitivity to different proxy metrics.


 -  While the paper includes several ablation studies, it does not provide a fine-grained analysis of the **Step-Wise Denoising Training (SDT)** component and the specific elements of the cache router. These elements are integral to HarmoniCa's efficacy, but without a breakdown of their contributions, it remains unclear how each sub-component (e.g., caching frequency, router sensitivity) directly impacts performance and efficiency.

- The paper provides comparisons with **Learning-to-Cache** and **FORA** but lacks discussion of other potential caching and acceleration techniques, such as model pruning or quantization, commonly explored in efficiency research for generative models. While these techniques are mentioned, a more in-depth comparison or consideration of combining methods could contextualize HarmoniCa’s strengths and limitations relative to other optimizations.

- Although HarmoniCa shows improvements in image quality (e.g., FID, IS), the paper could benefit from qualitative analyses of generated images. This would help illustrate how the framework’s focus on harmonizing training and inference affects visual output, potentially highlighting differences in quality between HarmoniCa and baseline methods.

**Questions:**

- **Question**: Could the authors elaborate on the criteria for selecting the error proxy in the Image Error Proxy-Guided Objective (IEPO)? Specifically, how was the proxy choice validated across different image generation tasks, and did the authors observe any trade-offs when experimenting with alternative metrics?

- **Question**: Could the authors provide additional insights into the computational resources required to implement HarmoniCa, especially the hardware and memory demands for training and inference?

- **Question**: Does HarmoniCa generalize well beyond the specific tasks demonstrated, such as high-resolution image generation? Have the authors considered exploring HarmoniCa’s effectiveness for video generation or other data-intensive applications where DiTs are applied?



- **Question**: How sensitive is HarmoniCa’s performance to the configuration of the cache router? Specifically, does adjusting parameters such as the caching threshold or the router’s update frequency significantly impact model performance, or is HarmoniCa robust across a range of these values?


- **Question**: Could the authors provide qualitative insights into the visual differences observed between HarmoniCa-generated images and those from baseline methods, especially in scenarios with aggressive caching? Are there specific artifacts or benefits visible that illustrate HarmoniCa’s impact on image quality?


- **Question**: Could HarmoniCa’s SDT and IEPO components be integrated into other popular architectures, such as U-Net-based diffusion models? If so, do the authors foresee any challenges or modifications needed to adapt HarmoniCa for such architectures?


- **Question**: Have the authors considered combining HarmoniCa with other model efficiency techniques, such as quantization or model pruning? What limitations or synergies do the authors anticipate in hybridizing HarmoniCa with these techniques?

**Details Of Ethics Concerns:**

There are no Ethics Concerns

---

> ### Author Response · Authors · 2024-11-17
> **Response to Reviewer UrSw (Part 1)**
>
> Thanks for the reviewer’s constructive comments.
>
> > **W1:** The HarmoniCa framework’s reliance on multiple components (SDT, IEPO, and the custom cache router) could limit its accessibility and reproducibility for practitioners, especially those with limited computational resources. While the model is described in detail, implementing and integrating SDT and IEPO into an existing Diffusion Transformer may pose a significant challenge due to the additional computational and architectural overhead.
> >
>
> Compared to the existing learning-based approach (i.e., Learning-to-Cache), HarmoniCa requires about **$\mathbf{\frac{3}{4}}$** of the training time without the need for any training images, making it a more user-friendly option. In terms of architecture, HarmoniCa does not require any modifications to the model itself. Furthermore, we only train a router which is a tensor with a shape of $T \times N$ (a few KB of memory), without any model weights. The minimal computation resource requirement for training with the gradient accumulation is under **12** GB of GPU memory for DiT-XL/2 256$\times$256, making it suitable even for an RTX 4070 Ti GPU. Moreover, users can employ gradient checkpointing to further reduce GPU memory consumption.
>
> > **W2:** While HarmoniCa is tested across multiple models and resolutions, it is evaluated only on **ImageNet** (for class-conditioned tasks) and **MS-COCO** (for text-to-image tasks). These datasets are popular benchmarks but may not represent the variety of distributions found in real-world data, especially in domains like video generation or multi-modal applications where diffusion models are increasingly applied.
> >
>
> In addition to the evaluations on ImageNet and MS-COCO, we conducted further tests using the high-quality MJHQ-30K [1] and sDCI [2] datasets with PixArt-$\alpha$ models. We added several metrics, including Image Reward [3], LPIPS [4], and PSNR. The results, summarized in the following table, demonstrate that HarmoniCa consistently outperforms DPM-Solver across all metrics on both the MJHQ and sDCI datasets. For instance, at the 512$\times$512 resolution, HarmoniCa achieves an FID of **7.01** on the MJHQ dataset, which is lower than the **7.04** FID of DPM-Solver with 20 steps, indicating better image quality. Additionally, under the same configuration, HarmoniCa achieves a PSNR of **22.09**, compared to DPM-Solver's **21.41** with 15 steps, reflecting better numerical similarity. Moving forward, we will explore the application of our method to video generation and multi-modal diffusion models in our future research. We have also included the results in Sec. Q.
>
> | Model | Method | T | MJHQ | MJHQ | MJHQ | MJHQ | MJHQ | sDCI | sDCI | sDCI | sDCI | sDCI | Latency (s)$\downarrow$ |
> | --- | --- | --- | --- | --- | --- | --- | --- | --- | --- | --- | --- | --- | --- |
> |  |  |  | Quality | Quality | Quality | Similarity | Similarity | Quality | Quality | Quality | Similarity | Similarity |  |
> |  |  |  | FID$\downarrow$   | IR$\uparrow$ | CLIP-Score$\uparrow$ | LPIPS$\downarrow$ | PSNR$\uparrow$ | FID$\downarrow$   | IR$\uparrow$ | CLIP-Score$\uparrow$ | LPIPS$\downarrow$ | PSNR$\uparrow$ |  |
> | 512$\times$512 | DPM-Solver | 20 | 7.04 | 0.947 | 26.04 | - | - | 11.47 | 0.994 | 25.22 | - | - | 1.759 |
> |  | DPM_Solver | 15 | 7.45 | 0.899 | 26.02 | 0.138 | 21.41 | 11.55 | 0.876 | 25.19 | 0.178 | 19.85 | 1.291 $_{(\times1.36)}$ |
> |  | HarmoniCa | 20 | **7.01** | **0.955** | **26.04** | **0.129** | **22.09** | **11.49** | **0.951** | **25.22** | **0.171** | **20.01** | **1.168 $_{(\times1.51)}$** |
> | 1024$\times$1024 | DPM-Solver | 20 | 6.24 | 0.966 | 26.23 | - | - | 10.96 | 0.986 | 25.56 | - | - | 9.470 |
> |  | DPM-Solver | 15 | 6.49 | 0.921 | 26.18 | 0.107 | 23.98 | 11.22 | 0.942 | 25.51 | 0.186 | 18.44 | 7.141 $_{(\times1.32)}$ |
> |  | HarmoniCa | 20 | **6.31** | **0.944** | **26.21** | **0.101** | **25.01** | **11.09** | **0.979** | **25.54** | **0.175** | **20.42** | **6.289 $_{(\times1.51)}$** |
>
> > **W3:** The **Image Error Proxy-Guided Objective (IEPO)** is a central contribution but relies on approximating final image quality based on intermediate denoising steps. The validity of this proxy could vary significantly depending on the task and model architecture, yet the paper provides limited empirical analysis of IEPO’s sensitivity to different proxy metrics.
> >
>
> Our superior performance and efficiency across class-conditional and text-to-image (T2I) tasks, evaluated on **8** models and **4** samplers with resolutions ranging from **256** $\times$ **256** to **2048** $\times$ **2048**, demonstrate the robustness and effectiveness of the current selected IEPO. Additionally, the strong results obtained across **5** different proxy metrics (as detailed in Sec. 5.3 and in response to Q1 from reviewer MqmJ) demonstrate that IEPO performs well with commonly used metrics such as MSE, MS-SSIM, and LPIPS for image reconstruction and quality assessment.

---

> ### Author Response · Authors · 2024-11-17
> **Response to Reviewer UrSw (Part 2)**
>
> > **W4:** While the paper includes several ablation studies, it does not provide a fine-grained analysis of the **Step-Wise Denoising Training (SDT)** component and the specific elements of the cache router. These elements are integral to HarmoniCa's efficacy, but without a breakdown of their contributions, it remains unclear how each sub-component (e.g., caching frequency, router sensitivity) directly impacts performance and efficiency.
> >
>
> We have provided an analysis of the Step-Wise Denoising Training (SDT) in Lines 295-297 included in our initial submission, where we analyze that SDT can reduce error at each timestep by incorporating information from prior timesteps during training. Additionally, our router is represented simply as a tensor with the shape $T \times N$, and does not involve any complex structures or design choices such as caching frequency or router sensitivity. Furthermore, we have presented a thorough ablation study in Sec. 5.3 in our initial submission. For instance, both SDT and IEPO achieve improvements exceeding **10** FID points over the baseline.
>
> > **W5:** The paper provides comparisons with **Learning-to-Cache** and **FORA** but lacks discussion of other potential caching and acceleration techniques, such as model pruning or quantization, commonly explored in efficiency research for generative models. While these techniques are mentioned, a more in-depth comparison or consideration of combining methods could contextualize HarmoniCa’s strengths and limitations relative to other optimizations.
> >
>
> Please refer to the answers to W1 and W2 from reviewer IBAN, where we compare our method with other caching approaches (DeepCache [5] and Faster Diffusion [6]), pruning methods (Diff-pruning [7]), and quantization techniques (PTQ4DiT [8] and EfficientDM [9]). All results validate the significant superiority of our method in both performance and efficiency. Furthermore, in Sec. I included in our initial submission, our method outperforms $\Delta$-DiT [10]. In Sec. F, the results demonstrate that combining quantization with our HarmoniCa method can achieve further acceleration without substantial performance degradation.
>
> > **W6:** Although HarmoniCa shows improvements in image quality (e.g., FID, IS), the paper could benefit from qualitative analyses of generated images. This would help illustrate how the framework’s focus on harmonizing training and inference affects visual output, potentially highlighting differences in quality between HarmoniCa and baseline methods.
> >
>
> We add the qualitative comparison and analyses in Sec. S. The more accurate details and objective-level traits help confirm HarmoniCa’s superiority over other baselines.
>
> > **Q1:** Could the authors elaborate on the criteria for selecting the error proxy in the Image Error Proxy-Guided Objective (IEPO)? Specifically, how was the proxy choice validated across different image generation tasks, and did the authors observe any trade-offs when experimenting with alternative metrics?
> >
>
> Please refer to the answer to the W3.
>
> > **Q2**: Could the authors provide additional insights into the computational resources required to implement HarmoniCa, especially the hardware and memory demands for training and inference?
> >
>
> Unlike methods such as pruning and quantization, HarmoniCa does not require any specialized hardware or custom CUDA kernel designs for training and inference. Taking DiT-XL/2 256$\times$256 as an example, the minimal memory demands are under **12** GB and approximately **4** GB of GPU memory for training and inference, respectively.
>
> > **Q3**: Does HarmoniCa generalize well beyond the specific tasks demonstrated, such as high-resolution image generation? Have the authors considered exploring HarmoniCa’s effectiveness for video generation or other data-intensive applications where DiTs are applied?
> >
>
> Yes, HarmoniCa generalizes well beyond the specific tasks demonstrated, including high-resolution image generation. For instance, in Sec. 5.2 and Sec. E included in our initial submission, we have shown that it performs effectively on 1024$\times$1024 PixArt-$\alpha$ and $\Sigma$, as well as 2048$\times$2048 PixArt-$\Sigma$. Specifically, it achieves over a **1.5** speedup ratio and a lower FID score for 1024$\times$1024 PixArt-$\alpha$ compared to the non-accelerated model. Additionally, we are actively exploring the application of our novel strategy to the video generation domain and other data-intensive tasks where DiTs are used.

---

> > ### Author Response · Authors · 2024-11-17
> > **Response to Reviewer UrSw (Part 3)**
> >
> > > **Q4**: How sensitive is HarmoniCa’s performance to the configuration of the cache router? Specifically, does adjusting parameters such as the caching threshold or the router’s update frequency significantly impact model performance, or is HarmoniCa robust across a range of these values?
> > >
> >
> > We conduct an ablation study on different values of the caching threshold $\tau \in [0,1)$, as shown in the table below. The results demonstrate that HarmoniCa is robust with variations of $\tau$. We have also included the results in Sec. R. Furthermore, the comprehensive ablation studies and the main results in the initial submission have further confirmed the generalizability and robustness of HarmoniCa for other hyper-parameters.
> >
> > | $\tau$ | T | IS$\uparrow$ | FID$\downarrow$ | sFID$\downarrow$ | Latency(s)$\uparrow$ |
> > | --- | --- | --- | --- | --- | --- |
> > | 0.1 | 10 | 151.83    | 13.35  | 11.13 | 0.270$_{(\times1.23)}$ |
> > | 0.5 | 10 | 151.80 | 13.41 | 11.09 | 0.269$_{(\times1.23)}$ |
> > | 0.9 | 10 | 151.78 | 13.37 | 11.08 | 0.270$_{(\times1.23)}$ |
> >
> > > **Q5**: Could the authors provide qualitative insights into the visual differences observed between HarmoniCa-generated images and those from baseline methods, especially in scenarios with aggressive caching? Are there specific artifacts or benefits visible that illustrate HarmoniCa’s impact on image quality?
> > >
> >
> > Please refer to the answer to the W6.
> >
> > > **Q6**: Could HarmoniCa’s SDT and IEPO components be integrated into other popular architectures, such as U-Net-based diffusion models? If so, do the authors foresee any challenges or modifications needed to adapt HarmoniCa for such architectures?
> > >
> >
> > Yes, HarmoniCa is architecture-agnostic and can be integrated into U-Net-based diffusion models. However, adapting HarmoniCa to other architectures may require careful consideration of the cache position, as some models have highly sensitive structures where caching could lead to significant performance degradation.
> >
> > > **Q7**: Have the authors considered combining HarmoniCa with other model efficiency techniques, such as quantization or model pruning? What limitations or synergies do the authors anticipate in hybridizing HarmoniCa with these techniques?
> > >
> >
> > Yes, we have explored combining HarmoniCa with quantization techniques, such as EfficientDM [7], as discussed in Sec. F included in the initial submission. By combining these two orthogonal methods, we achieve further speedup, though at the cost of some performance degradation. We plan to investigate the optimal combination of these techniques in future research.
> >
> > [1] Li D, Kamko A, Akhgari E, et al. Playground v2. 5: Three insights towards enhancing aesthetic quality in text-to-image generation[J]. arXiv preprint arXiv:2402.17245, 2024.
> >
> > [2] Urbanek J, Bordes F, Astolfi P, et al. A picture is worth more than 77 text tokens: Evaluating clip-style models on dense captions[C]//Proceedings of the IEEE/CVF Conference on Computer Vision and Pattern Recognition. 2024: 26700-26709.
> >
> > [3] Xu J, Liu X, Wu Y, et al. Imagereward: Learning and evaluating human preferences for text-to-image generation[J]. Advances in Neural Information Processing Systems, 2024, 36.
> >
> > [4] Zhang R, Isola P, Efros A A, et al. The unreasonable effectiveness of deep features as a perceptual metric[C]//Proceedings of the IEEE conference on computer vision and pattern recognition. 2018: 586-595.
> >
> > [5] Ma X, Fang G, Wang X. Deepcache: Accelerating diffusion models for free[C]//Proceedings of the IEEE/CVF Conference on Computer Vision and Pattern Recognition. 2024: 15762-15772.
> >
> > [6] Li S, Hu T, Shahbaz Khan F, et al. Faster diffusion: Rethinking the role of unet encoder in diffusion models[J]. arXiv e-prints, 2023: arXiv: 2312.09608.
> >
> > [7] He Y, Liu J, Wu W, et al. Efficientdm: Efficient quantization-aware fine-tuning of low-bit diffusion models[J]. arXiv preprint arXiv:2310.03270, 2023.
> >
> > [8] Wu J, Wang H, Shang Y, et al. PTQ4DiT: Post-training Quantization for Diffusion Transformers[J]. arXiv preprint arXiv:2405.16005, 2024.
> >
> > [9] Wang Z, Jiang Y, Zheng H, et al. Patch diffusion: Faster and more data-efficient training of diffusion models[J]. Advances in neural information processing systems, 2024, 36.
> >
> > [10] Chen P, Shen M, Ye P, et al. $\Delta$-DiT: A Training-Free Acceleration Method Tailored for Diffusion Transformers[J]. arXiv preprint arXiv:2406.01125, 2024.

---

> > > ### Comment · Reviewer_UrSw · 2024-11-26
> > >
> > > Thank you for your response, the issue I was concerned about has been resolved. I recommend accepting this article.

---

> > > > ### Author Response · Authors · 2024-11-26
> > > > **Thanks for your feedback**
> > > >
> > > > Dear Reviewer UrSw,
> > > >
> > > > Thank you for your feedback! We appreciate the constructive reviews for improving our work.
> > > >
> > > > Best regards,
> > > >
> > > > Authors of Paper #1567

---

### Official Review · Reviewer_Ukn2 · 2024-11-03

**Soundness:** 3
**Presentation:** 3
**Contribution:** 2
**Rating:** 5
**Confidence:** 4

**Summary:**

This paper introduces HarmoniCa, a learnable caching method designed for the Diffusion Transformer architecture. It primarily addresses the issues of Prior Timestep Disregard and Objective Mismatch found in previous learnable caching methods.

**Strengths:**

1.	This paper identifies issues of Prior Timestep Disregard and Objective Mismatch in current learning-based feature caching methods and proposes SDT and IEPO to address them, respectively.
2.	Experiments on the DiT-XL/2, PixART-α, and PIXART-Σ families, including eight models, four samplers, and four resolutions, demonstrate the effectiveness and generality of harmonica.
3.	The paper is well organized and clearly presented.

**Weaknesses:**

1. This work appears to be an incremental improvement on Learn-to-Cache, which initially made caching mechanisms learnable. The main contribution here is refining the learning mechanism of Learn-to-Cache. However, in terms of both efficiency and generation metrics, HarmoniCa shows only limited improvements.
2. In Table 2, HarmoniCa’s results are only slightly better than those of Learn-to-Cache. In Table 3, under similar latency and computational requirements, HarmoniCa’s generation metrics are nearly the same as those of SA-Solver, a method that requires no training. Thus, the need for training in HarmoniCa does not offer a clear advantage, with similar observations across other solvers.

**Questions:**

Please refer to the weaknesses part.

---

> ### Author Response · Authors · 2024-11-17
> **Response to Reviewer UKn2 (Part 1)**
>
> Thanks for the reviewer’s constructive comments.
>
> > **W1:** This work appears to be an incremental improvement on Learn-to-Cache, which initially made caching mechanisms learnable. The main contribution here is refining the learning mechanism of Learn-to-Cache. However, in terms of both efficiency and generation metrics, HarmoniCa shows only limited improvements.
> >
>
> While our work builds upon the foundational concept (i.e., learn a caching strategy) introduced by Learn-to-Cache, our contributions and improvements are both significant and distinct in the following ways:
>
> 1. **Identifying Key Problems**: We first identify two critical discrepancies between training and inference overlooked in Learn-to-Cache as follows：
>     1. **Prior timestep disregard**: Ignore the error caused by reusing cached features at prior timesteps during optimization.
>     2. **Objective Mismatch:** Focus on optimizing the intermediate output during the denoising process.
>
>     They both significantly affect the overall performance of Learning-to-Cache.
>
> 2. **Novel Training Strategies**: To address these problems, we propose two novel training techniques (as recognized by reviewers **zBAN** and **UrSw**) (i.e., SDT+IEPO). **Our entire optimization processes differ fundamentally** compared with Learning-to-Cache:
>     1. For the training paradigm, Learning-to-Cache uses random timestep sampling similar to DDPM, focusing solely on the current sampled timestep. In contrast, we implement SDT for iterative denoising from random noise, which accounts for all previous timesteps, mirroring the inference process.
>     2. For the learning objective, Learning-to-Cache focuses on intermediate noise and cache usage. We further optimize the final image error and cache usage with our IEPO.
>
>     The proposed framework achieves significantly improved performance and more efficient caching (as highlighted by reviewers **zBAN**, **MqmJ**, and **UrSw)**.
>
> 3. **Reduced Training Cost**: Our method significantly reduces training costs. Compared to Learning-to-Cache, it requires significantly **less training time** (i.e., $\mathbf{\frac{3}{4}}$ of the training time) **without any training images**, making it more user-friendly and easier to implement. This is especially advantageous for advanced generative models (e.g., PixArt-$\alpha/\Sigma$), where applying Learn-to-Cache is less feasible (e.g., more training time and large pre-train datasets are required).
> 4. **Substantial Performance Improvements:**
>     1. HarmoniCa consistently and pronouncedly outperforms Learn-to-Cache across all settings, particularly when fewer steps are used for sampling. For instance, in Tab. 2, HarmoniCa achieves a **1.24** improvement in FID and a **6.74** improvement in IS compared to Learn-to-Cache with the 10-step DiT-XL/2, under a higher speedup ratio.
>     2. We also **add IS and FID results across different speedup ratios in Sec. O**, showing that our method yields significantly better results as the speedup ratio increases, e.g., **30.90** higher IS and **12.34** lower FID scores at about $\times$1.6 speedup. HarmoniCa handles the trend towards fewer timesteps and higher speedup ratios far more effectively than Learn-to-Cache.

---

> > ### Author Response · Authors · 2024-11-17
> > **Response to Reviewer UKn2 (Part 2)**
> >
> > > **W2:** In Table 2, HarmoniCa’s results are only slightly better than those of Learn-to-Cache. In Table 3, under similar latency and computational requirements, HarmoniCa’s generation metrics are nearly the same as those of SA-Solver, a method that requires no training. Thus, the need for training in HarmoniCa does not offer a clear advantage, with similar observations across other solvers.
> > >
> >
> > In response to the first question, our HarmoniCa demonstrates a significant improvement over Learning-to-Cache. Regarding the comparison with SA-Solver, we conducted additional experiments to highlight HarmoniCa's advantages.
> >
> > In the first table below, we use fewer denoising steps (20 steps, compared to 25 in the paper). With similar latency, our method outperforms the 16-step SA-Solver by **2.10** FID and **3.3** sFID. In the second table, we test our method with higher resolutions. As resolution increases, HarmoniCa delivers more pronounced benefits than the fewer-step SA-Solver. Specifically, HarmoniCa achieves lower FID and sFID, and a higher CLIP score with a **1.46$\times$** speedup over the non-accelerated model. In contrast, the 20-step SA-Solver performs worse than the non-accelerated model, with a **1.30$\times$** speedup. We have also included the results in Sec. P.
> >
> > For the other samplers in the initial submission (Tab. 2), all of them using HarmoniCa achieve more than **1.5**$\times$ speedup, while the fewer-step samplers achieve less than **1.4$\times$** speedup and consistently worse performance across all metrics.
> >
> > | Method | T | CLIP$\uparrow$ | FID$\downarrow$ | sFID$\downarrow$ | CUR(%)$\uparrow$ | Latency(s)$\downarrow$ |
> > | --- | --- | --- | --- | --- | --- | --- |
> > | PixArt-$\alpha$  256$\times$﻿256 |  |  |  |  |  |  |
> > | SA-Solver | 20 | 31.28 | 23.96 | 35.63 | - | 0.677 |
> > | SA-Solver | 16 | 31.16 | 26.27 | 39.28 | - | 0.520$_{(\times1.30)}$ |
> > | HarmoniCa | 20 | **31.23** | **24.17** | **35.98** | **42.12** | **0.516**$_{(\times1.31)}​$ |
> >
> > | Method | T | CLIP$\uparrow$ | FID$\downarrow$ | sFID$\downarrow$ | CUR(%)$\uparrow$ | Latency(s)$\downarrow$ |
> > | --- | --- | --- | --- | --- | --- | --- |
> > | PixArt-$\alpha$ 512$\times$512 |  |  |  |  |  |  |
> > | SA-Solver | 25 | 31.23 | 25.43 | 39.84 | - | 2.263 |
> > | SA-Solver | 20 | 31.19 | 25.85 | 40.08 | - | 1.738$_{(\times1.30)}$ |
> > | HarmoniCa | 25 | **31.24** | **24.44** | **39.87** | **52.04** | **1.611$_{(\times1.40)}$** |
> > | PixArt-$\alpha$ 1024$\times$1024 |  |  |  |  |  |  |
> > | SA-Solver | 25 | 31.05 | 23.65 | 38.12 | - | 11.931 |
> > | SA-Solver | 20 | 31.02 | 23.88 | 39.41 | - | 9.209$_{(\times1.30)}$ |
> > | Harmonica | 25 | **31.10** | **23.52** | **37.89** | **52.46** | **8.151$_{(\times1.46)}$** |

---

> > > ### Author Response · Authors · 2024-11-23
> > > **Reaching the End of the Public Discussion Phase**
> > >
> > > Dear Reviewer,
> > >
> > > Thank you for handling our manuscript and providing valuable feedback. We hope that our responses have sufficiently addressed the concerns you raised. We welcome more discussion if you have more questions and suggestions. As the discussion deadline is approaching, we would be very grateful if you could take a moment to review our reply.

---

> ### Author Response · Authors · 2024-11-25
> **Reaching the End of the Public Discussion Phase (Only Two Days Left)**
>
> Dear Reviewer,
>
> Sorry for bothering you again, but the discussion period is coming to an end in two days. Could you please let us know if our responses have alleviated your concerns? If there are any further comments, we will do our best to respond.
>
> Best regards,
>
> Authors of Paper #1567

---

> ### Author Response · Authors · 2024-11-26
> **Final Request for Feedback Before PDF Submission Deadline**
>
> Dear Reviewer,
>
> We hope this message finds you well. I apologize for the repeated follow-ups, but as the deadline to submit the revised PDF is tomorrow, we want to kindly remind you that we have provided additional clarifications and revisions in response to your comments. If there are any remaining concerns or further points you would like us to address, we would be very grateful for your feedback and would be happy to make any necessary adjustments.
>
> We truly appreciate your time and thoughtful consideration in reviewing our work.
>
> Thank you so much for your understanding and support.
>
> Best regards,
>
> Authors of Paper #1567

---

> ### Author Response · Authors · 2024-11-28
> **Follow-up on Paper #1567 After PDF Submission Deadline**
>
> Dear Reviewer,
>
> We hope this message finds you well. Apologies for following up again, but as the discussion period is coming to a close, we wanted to kindly check if our responses have sufficiently addressed your concerns. We have carefully revised the manuscript based on your feedback and have provided further clarifications in the comment section.
>
> If you have any additional points or questions, we would be happy to address them. We truly appreciate your time and thoughtful consideration, and we are grateful for the opportunity to improve the manuscript with your feedback.
>
> If you're celebrating, we would like to wish you a joyful and restful Thanksgiving!
>
> Thank you once again for your continued support.
>
> Best regards,
>
> Authors of Paper #1567

---

> ### Author Response · Authors · 2024-12-01
> **Urgent Follow-up to Reviewer Ukn2 – Discussion Deadline Approaching**
>
> Dear Reviewer Ukn2,
>
> We hope this message finds you well. We apologize for reaching out repeatedly, but with **less than two days remaining in the discussion period**, we wanted to check whether our updates have addressed your concerns kindly.
>
> If you have any further questions or require additional clarification, we are more than happy to provide it. Your feedback has been invaluable, and we greatly appreciate your time and effort in reviewing our work.
>
> Thank you again for your continued support. We look forward to your feedback.
>
> Best regards,
>
> Authors of Paper #1567

---

> ### Author Response · Authors · 2024-12-02
> **Warm Appreciation and Invitation for Further Feedback – Discussion Left Less than 1 Day**
>
> Dear Reviewer Ukn2,
>
> We apologize for reaching out repeatedly again. We would like to sincerely thank you for your thoughtful review and valuable feedback on our paper. We wanted to kindly check if our responses have sufficiently addressed your concerns. We have carefully revised the manuscript based on your feedback and have provided further clarifications in the comment section.
>
> If you find that our revisions have satisfactorily addressed your concerns, we would be grateful if you could consider reflecting this in your final assessment.
>
> Thank you again for your valuable contributions to our research.
>
> Warm regards,
>
> Authors of Paper #1567

---

> ### Author Response · Authors · 2024-12-03
> **Warm Appreciation and Invitation for Further Feedback – Discussion Left Less than a Half Day**
>
> Dear Reviewer,
>
> We hope this message finds you well. We apologize for the repeated follow-ups and want to kindly remind you that we have provided additional clarifications and revisions in response to your comments. If there are any remaining concerns or further points you would like us to address, we would be very grateful for your feedback and would be happy to make any necessary adjustments.
>
> We truly appreciate your time and thoughtful consideration in reviewing our work.
>
> Thank you so much for your understanding and support.
>
> Best regards,
>
> Authors of Paper #1567

---

### Official Review · Reviewer_MqmJ · 2024-11-04

**Soundness:** 3
**Presentation:** 2
**Contribution:** 2
**Rating:** 6
**Confidence:** 3

**Summary:**

This paper introduces Harmonica, a learning-based caching method that accelerates the sampling of the diffusion model by reusing the neural network features. Harmonica improves the training algorithm and training objective for the learnable router, building upon the Learning-to-cache framework. Extensive experiments on ImageNet and MS-COCO demonstrate the sampling efficiency of Harmonica across different models and resolutions.

**Strengths:**

1. The proposed method, Harmonica, is clean and effective with strong empirical performance.
2. Harmonica eliminates the need for external datasets, unlike the existing learning-based method.
3. Demonstrates practical benefits across different architectures and scales.
4. Comprehensive ablation study.

**Weaknesses:**

1. Harmonica introduces additional hyperparameters, such as the coefficient $ beta$ and update interval $ C$, which require more computation to tune. While this additional offline cost does not affect the inference speedup, the paper should still provide the cost spent on tuning the hyperparameters to help readers better understand the full implementation costs.
2. Overall, the writing could be largely improved by breaking the long sentences into shorter ones. Here are some typos I caught :
	1. Line 127, "much lower training costs" -> "much lower training cost".
	2. Line 213, "Harmonize" -> "harmonize".
	3. Line 261, "The student" -> "the student".
	4. The second DiT block of the bottom row has the same format as Teacher DiT in Figure 4 (b).
3. Terminology: "discrepancy" could be replaced with "mismatch" (as used in line 237) for better accuracy as discrepancy often refers to a lack of similarity between quantities.

**Questions:**

1. The authors explore different standard metrics for $\lambda (t)$, but none are designed for natural images. Would perceptual metrics like LPIPS be more effective in measuring image similarity?
2. Why is $\lambda (t)$ designed to be multiplied with $L_{MSE}$ ? Are there any other alternative design choices there?
3. Can the router trained with one diffusion sampler be applied to a different diffusion sampler? For example, the training uses a DDIM sampler, but the sampling uses a DPM solver.
4. Implementation details:
	1. Is $r_{t,i}$ implemented as an output of sigmoid function following Learning-to-Cache?
	2. Is $r_{r,i}$ updated using the vanilla gradient descent or other optimizers?
	3. Does Harmonica apply the stop gradient to `gen_proxy`? In other words, $\lambda (t)$ is an adaptive weight of $L_{MSE}$.

---

> ### Author Response · Authors · 2024-11-17
> **Response to Reviewer MqmJ (Part 1)**
>
> Thanks for the reviewer’s constructive comments.
>
> > **W1:** Harmonica introduces additional hyperparameters, such as the coefficient  $\beta$ and $C$update interval , which require more computation to tune. While this additional offline cost does not affect the inference speedup, the paper should still provide the cost spent on tuning the hyperparameters to help readers better understand the full implementation costs.
> >
>
> We adopt a grid search approach by sampling several values for $C$ and benchmarking feature cache performance (as shown in Fig. 7 in the initial submission). Based on this, we fix $C$ as 500 for all experiments. The improved performance and acceleration ratios validate the robustness of this fixed value which can be directly applied to other settings, making **the overhead a one-time cost** (about $5\times8$ H800-GPU-hours). For $\beta$, initially, we tested multiple values and found that they exhibited robustness across different configurations for a model series. Therefore, once an effective range is established, we can get a satisfactory $\beta$ with just a few trials (i.e., a few iterations). On average, determining $\beta$ takes just **over 10 minutes per setting**, making the process efficient and practical. In future work, we plan to explore tuning-based approaches to automate the determination of these hyperparameters.
>
> > **W2:** Overall, the writing could be largely improved by breaking the long sentences into shorter ones. Here are some typos I caught :
> >
> > 1. Line 127, "much lower training costs" -> "much lower training cost".
> > 2. Line 213, "Harmonize" -> "harmonize".
> > 3. Line 261, "The student" -> "the student".
> > 4. The second DiT block of the bottom row has the same format as Teacher DiT in Figure 4 (b).
>
> We sincerely thank the reviewer for providing valuable suggestions and pointing out these. We have fixed these typos and broken the long sentences in the paper.
>
> > **W3:** Terminology: "discrepancy" could be replaced with "mismatch" (as used in line 237) for better accuracy as discrepancy often refers to a lack of similarity between quantities.
> >
>
> We sincerely appreciate the reviewer’s thoughtful suggestion and have carefully reviewed this point. After thorough consideration, the term “discrepancy” has been retained, as it is widely used in prior studies (e.g., [1, 2]) to describe the inconsistency between training and inference. Retaining this term ensures alignment with established terminology in the field and avoids potential overlap with “Objective Mismatch.” To further address the reviewer’s concern, the definition of “discrepancy” has been clarified in Lines 52-53 of the revised manuscript. The authors hope this revision adequately addresses the reviewer’s feedback.
>
> > **Q1:** The authors explore different standard metrics for $\lambda^{(t)}$, but none are designed for natural images. Would perceptual metrics like LPIPS be more effective in measuring image similarity?
> >
>
> Thanks for the insightful suggestion. Under the same speedup ratio, we test $\text{MS-SSIM}$ and $\text{LPIPS}$ as suggested. In the table below, these metrics exhibit comparable performance compared with$\|\|x_0-x_0^{(t)}\|\|_F^2$. For instance, $\text{LPIPS}(x_0, x_0^{(t)})$ slightly outperforms in FID and sFID, while $\|\|x_0-x_0^{(t)}\|\|_F^2$ marginally excels in IS. We have included the results in Sec. M and will explore more about the influence of metric selection in the future.
>
> | $\lambda^{(t)}$ | $\|\|x_0-x_0^{(t)}\|\|_F^2$ | $1-\text{MS-SSIM}(x_0, x_0^{(t)})$ | $\text{LPIPS}(x_0, x_0^{(t)})$ |
> | --- | --- | --- | --- |
> | IS$\uparrow$ | **206.57** | 204.72 | 205.83 |
> | FID$\downarrow$ | 4.88 | 4.91 | **4.83** |
> | sFID$\downarrow$ | 5.91 | 5.83 | **5.57** |
> | CUR(%)$\uparrow$ | 37.50 | **37.68** | 37.32 |
> | Latency$\downarrow$ | **0.456$_{(\times1.44)}$** | **0.456$_{(\times1.44)}$** | **0.456$_{(\times1.44)}$** |
>
> > **Q2:** Why is  $\lambda^{(t)}$ designed to be multiplied with $\mathcal{L}_{MSE}$? Are there any other alternative design choices there?
> >
>
> Since our goal is to use image error to guide cache usage (i.e., tend to reuse fewer features if the final error is large), $\lambda^{(t)}$, which reflects the final image error, serves as a multiplier to control the weight of $\mathcal{L}\_{MSE}$ in the loss function. A larger $\lambda^{(t)}$ (indicating a larger error) increases the emphasis on reducing $\mathcal{L}\_{MSE}$, encouraging less feature reuse, and vice versa. These explanations can be found in Lines 310-319. Using $\lambda^{(t)}$ as a simple multiplier is effective, but considering it as an exponent of $\mathcal{L}\_{MSE}$ or exploring other algebraic forms offers valuable, albeit more complex, alternatives. We plan to investigate these designs in our future research.

---

> ### Author Response · Authors · 2024-11-17
> **Response to Reviewer MqmJ (Part 2)**
>
> > **Q3:** Can the router trained with one diffusion sampler be applied to a different diffusion sampler? For example, the training uses a DDIM sampler, but the sampling uses a DPM solver.
> >
>
> As shown in the table below, the router trained with one diffusion sampler can indeed be applied to a different sampler, such as DPM-Solver++$\rightarrow$SA-Solver (5th row) and IDDPM$\rightarrow$DPM-Solver++ (9th row). However, the performance of these trials is much worse than the standard HarmoniCa. We believe this is due to the discrepancies in sampling trajectories and noise scheduling between the two samplers, which need to be accounted for during router training. In other words, the sampler used for training should match the one used during inference to improve the performance. We have also included the results and relevant discussions in Sec. N.
>
> | Method | T | CLIP$\uparrow$ | FID$\downarrow$ | sFID$\downarrow$ | CUR(%)$\uparrow$ | Latency(s)$\downarrow$ |
> | --- | --- | --- | --- | --- | --- | --- |
> | PixArt-$\alpha$ 256$\times$256 |  |  |  |  |  |  |
> | SA-Solver | 20 | 31.28 | 23.96 | 35.63 | - | 0.677 |
> | SA-Solver | 16 | 31.16 | 26.27 | 39.28 | - | 0.520$_{(\times1.30)}$ |
> | HarmoniCa | 20 | **31.23** | **24.17** | **35.98** | **42.12** | **0.516$_{(\times1.31)}$** |
> | HarmoniCa (DPM-Solver++$\rightarrow$SA-Solver) | 20 | 31.18 | 25.99 | 37.94 | 40.98 | 0.523$_{(\times1.29)}$ |
> | DPM-Solver++ | 100 | 31.30 | 25.01 | 35.42 | - | 2.701 |
> | DPM-Solver++ | 73 | 31.27  | 25.16 | 36.11 | - | 2.005$_{(\times1.35)}$ |
> | HarmoniCa | 100 | **31.35** | **24.96** | **35.19** | **51.89** | **1.998$_{(\times1.35)}$** |
> | HarmoniCa (IDDPM$\rightarrow$DPM-Solver++) | 100 | 31.22 | 25.43 | 39.84 | 50.98 | 2.002$_{(\times1.35)}$ |
>
> > **Q4:** Implementation details:
> >
> > 1. Is  $r_{t,i}$ implemented as an output of sigmoid function following Learning-to-Cache?
> > 2. Is  $r_{t,i}$ updated using the vanilla gradient descent or other optimizers?
> > 3. Does Harmonica apply the stop gradient to `gen_proxy`? In other words,  $\lambda^{(t)}$ is an adaptive weight of $\mathcal{L}\_{MSE}$.
> 1. Yes (as mentioned in Sec. D).
> 2. We employ AdamW optimizer with the default setting in PyTorch to update the value (as mentioned in Sec. 5.1).
> 3. Yes, we directly employ `with torch.no_grad():` to wrap `gen_proxy`.
>
> [1] Ning M, Sangineto E, Porrello A, et al. Input perturbation reduces exposure bias in diffusion models[J]. arXiv preprint arXiv:2301.11706, 2023.
>
> [2] Li M, Qu T, Yao R, et al. Alleviating exposure bias in diffusion models through sampling with shifted time steps[J]. arXiv preprint arXiv:2305.15583, 2023.

---

### Official Review · Reviewer_zBAN · 2024-11-05

**Soundness:** 2
**Presentation:** 3
**Contribution:** 2
**Rating:** 6
**Confidence:** 4

**Summary:**

This paper proposes a new framework, HarmoniCa, which is a learning-based cache mechanism. The authors discover two main issues with previous methods: prior timestep disregard and objective mismatch. Step-wise Denoising Training (SDT) and Image Error-Aware Optimization Objective (IEPO) are proposed to address these two issues. Experiments show that the proposed framework can achieve a better performance and efficiency with lower training costs.

**Strengths:**

1. This paper is clearly written and easy to follow, presenting a new learning-based Caching framework that can address the two discrepancies between training and inference identified by the authors.
2. Extensive experiments on different models manifest the efficacy and universality of HarmoniCa with much lower training costs.

**Weaknesses:**

1. This paper lacks experimental comparisons with other cache-based methods, such as DeepCache [R1] and Faster Diffusion [R2].
2. A performance comparison with other related methods under the same acceleration ratio, such as pruning or quantization, should be provided for further validation.

[R1] Ma X, Fang G, Wang X. Deepcache: Accelerating diffusion models for free[C]//Proceedings of the IEEE/CVF Conference on Computer Vision and Pattern Recognition. 2024: 15762-15772.

[R2] Li S, Hu T, Shahbaz Khan F, et al. Faster diffusion: Rethinking the role of unet encoder in diffusion models[J]. arXiv e-prints, 2023: arXiv: 2312.09608.

**Questions:**

The questions are similar to those mentioned in weaknesses.

1. Can the authors provide more comprehensive comparison results with other cache-based models?
2. What are the comparative results of other related methods under the same acceleration ratio?

---

> ### Author Response · Authors · 2024-11-17
> **Response to Reviewer zBAN**
>
> Thanks for the reviewer’s constructive comments.
>
> > **W1:** This paper lacks experimental comparisons with other cache-based methods, such as DeepCache and Faster Diffusion.
> >
>
> To highlight HarmoniCa's advantages, we compare it with DeepCache and Faster Diffusion. Notably, DeepCache is constrained by the U-shaped structure, making it unsuitable for DiTs. Since our approach is not constrained by any specific structure, we use the U-Vit[1] model with the ImageNet 256$\times$256 dataset on a single A6000 GPU. We have also included the results and experimental details in Sec. K. As shown in the table, HarmoniCa achieves a minimal FID increase of less than **0.05**, while providing a **1.65$\times$** speedup, outperforming both Faster Diffusion and DeepCache.
>
> | Methods | T | FID$\downarrow$ | Latency(s)$\downarrow$ |
> | --- | --- | --- | --- |
> | DPM-Solver | 20 | 2.57 | 7.60 |
> | Faster Diffusion | 20 | 2.82 | 5.95$_{(\times1.28)}$ |
> | DeepCache | 20 | 2.70 | 4.68$_{(\times1.62)}$ |
> | HarmoniCa | 20 | **2.61** | **4.60$_{(\times1.65)}$** |
>
> > **W2:** A performance comparison with other related methods under the same acceleration ratio, such as pruning or quantization, should be provided for further validation.
> >
>
> As shown in the table below, we compare our HarmoniCa with advanced quantization (w8a8) and pruning methods using DiT-XL/2 on ImageNet 256$\times$256. We have also included the results and experimental details in Sec. L. Our method significantly outperforms other methods, demonstrating the substantial benefit of feature cache for accelerating DiT models. It is important to note that the speedup ratio for quantization is partially determined by hardware support, which we do not rely on. Additionally, our method is orthogonal to these approaches, meaning it can be combined with them for further acceleration, e.g., **1.80**$\times$speedup on H800 (results of EfficientDM + HarmoniCa have been presented in Sec. F included in our initial submission).
>
> | Methods | T | IS$\uparrow$ | FID$\downarrow$ | sFID$\downarrow$ | Latency(s)$\downarrow$（H800） | Latency(s)$\downarrow$ (A100) |
> | --- | --- | --- | --- | --- | --- | --- |
> | DDIM | 20 | 224.37 | 3.52 | 4.96 | 0.658 | 1.217 |
> | EfficientDM [2] | 20 | 172.70 | 6.10 | **4.55** | 0.591$_{(\times1.11)}$ | 0.842$_{(\times1.45)}$ |
> | PTQ4DIT [3] | 20 | 17.06 | 71.82 | 23.16 | ​0.577$_{(\times1.14)}$ | 0.839$_{(\times1.45)}$ |
> | Diff-pruning [4] | 20 | 168.10 | 8.22 | 6.20 | 0.458$_{(\times1.44)}$ | **0.813$_{(\times1.50)}$** |
> | HarmoniCa | 20 | **206.57** | **4.88** | 5.91 | **0.456**$_{(\times1.44)}$ | 0.815$_{(\times1.49)}$ |
>
> > **Q1:** Can the authors provide more comprehensive comparison results with other cache-based models?
> >
>
> Please refer to the answer to the W1.
>
> > **Q2:** What are the comparative results of other related methods under the same acceleration ratio?
> >
>
> Please refer to the answer to the W2.
>
> [1] Bao F, Nie S, Xue K, et al. All are worth words: A vit backbone for diffusion models[C]//Proceedings of the IEEE/CVF conference on computer vision and pattern recognition. 2023: 22669-22679.
>
> [2] He Y, Liu J, Wu W, et al. Efficientdm: Efficient quantization-aware fine-tuning of low-bit diffusion models[J]. arXiv preprint arXiv:2310.03270, 2023.
>
> [3] Wu J, Wang H, Shang Y, et al. PTQ4DiT: Post-training Quantization for Diffusion Transformers[J]. arXiv preprint arXiv:2405.16005, 2024.
>
> [4] Fang, Gongfan et al. “Structural Pruning for Diffusion Models.” *ArXiv* abs/2305.10924 (2023): n. pag.

---

### Author Response · Authors · 2024-11-17
**Response to all Reviewers and Area Chairs**

Dear Reviewers and  Area Chairs,

We sincerely thank all reviewers and the area chair for their insightful comments and valuable time. We are grateful for the positive recognition of the strengths of our work. The reviewers agree that:

**The Novel Approach:**

- *"… presenting a new learning-based caching framework that can address the two discrepancies…"* **(Reviewer zBAN)**
- *“HarmoniCa addresses this through a novel feature caching framework…’’* **(Reviewer UrSw)**

**Strong Practical Contributions:**

- *“… proposed framework can achieve a better performance and efficiency with lower training costs.”* **(Reviewer zBAN)**
- *"HarmoniCa eliminates the need for external datasets ... Demonstrates practical benefits across different architectures and scales."*  and “… *HarmoniCa, is clean and effective with strong empirical performance.*”**(Reviewer MqmJ)**
- *"The HarmoniCa framework represents a substantial step forward in making high-resolution generative models more practical by lowering inference and training costs."* **(Reviewer UrSw)**

**Comprehensive Experiments:**

- “Extensive experiments on different models manifest …” **(Reviewer zBAN)**
- “Extensive experiments on ImageNet and …” **(Reviewer MqmJ)**
- *"Experiments on the DiT-XL/2, PixART-α, and PIXART-Σ families, including eight models, four samplers, and four resolutions ..."* **(Reviewer Ukn2)**
- *"The authors provide comprehensive experiments across ..."* **(Reviewer UrSw)**

We have carefully addressed all of your comments and suggestions, and we hope that our revisions have strengthened the paper further. Thank you again for your valuable feedback and support.

---

### Meta-Review · Area_Chair_4zzR · 2024-12-20

**Metareview:**

The paper proposes a learning based cache mechanism for diffusion models.
The mechanism involves storing and retrieving redundant computations across timesteps.
The method can slightly improve latency (about 1.5x) and training costs (25% improvement).

* Reviewer zBAN (6) find the paper clearly written and the experiments extensive. However, experiment comparisons to related methods (such as DeepCache) were missing in the original paper. In the rebuttal, the authors included those comparisons, and the reviewer notes that most of their concerns are addressed.

* Reviewer MqmJ (6) notes that the method is effective and performs well, but thinks that that writing could be improved significantly.

* Reviewer Ukn2 (5) notes that the improvement over learn-to-cache is minor, conceptually, and in performance and efficiency. The authors argue that their contribution is to identify two issues of learn-to-catch (prior timestep disregard and objective mismatch) and addressing those makes their Harmonica method a better algorithm.

* Reviewer UrSW (8) provides a detailed review but notes that they are unable to assess the paper. After careful reading and consideration, this review was not included into the final evaluation.

The paper received somewhat mixed reviews and reviewers expressed uncertainty in their certainty about their evaluation. I agree with the authors that their method improves over learn-to-cache, but I also agree with reviewer Ukn2 that the improvement over existing methods is relatively minor, and the results are somewhat overstated. Based on my own reading, I found parts difficult to follow, and no code for reproducibility is provided.

**Additional Comments On Reviewer Discussion:**

please see meta review

---

### Decision · Program_Chairs · 2025-01-22

Reject